# Attention-Head Binding as a Term-Conditioned Mechanistic Marker of Accessibility Concept Emergence in Language Models

**Khanh-Dung Tran**  ⓘ                                 *dung@therayo.com*
*Rayo Lab Pte. Ltd., Singapore*

**Reviewed on OpenReview:** *https://openreview.net/forum?id=QG7mfCy9mu*

## Abstract

Assessing when language models develop specific capabilities remains challenging, as behavioral evaluations are expensive and internal representations are opaque. We introduce *attention-head binding* (EB*), a lightweight mechanistic metric that tracks how attention heads bind multi-token technical terms, such as accessibility concepts ("screen reader," "alt text"), into coherent units during training. Using **seven models** across **five architectures**, including Pythia (160M, 1B, 2.8B), OLMo-1B, CRFM GPT-2 Small (5 seeds), SmolLM3-3B, and Qwen2.5-1.5B, we evaluate on **41 canonical accessibility terms** (N=205 prompts) and the 9-term pilot set, reporting five empirical findings. Discriminant validity validates EB* against token co-occurrence baselines (nonsense $0.26 \rightarrow$ real terms 0.74, all $p < 0.001$, $d = 1.2$–2.9). The relationship between binding and behavior shifts markedly over the course of training. Early in training, the two are tightly coupled ($\rho = +0.57$, $p < 0.001$). Later, this pattern reverses into a decoupled regime ($\rho = -0.20$, $p = 0.01$). **Cross-architecture replication** confirms C1-B: OLMo-1B achieves 90% EB*-leads ($p < 0.0001$), CRFM 72.7% ($p \ll 0.001$). This gives rise to a two-factor model. First, a parameter threshold around 1B parameters controls how deeply decoupling occurs. Second, a training-step threshold near 300K steps determines when the temporal ordering between binding and behavior emerges (C1/C4). High-binding/mid-accuracy checkpoints contain unlockable latent knowledge, yielding few-shot gains up to 61 percentage points (a 183% relative improvement), replicated at 18–37 points across six of seven models (CRFM shows weak unlockability at +7.6 pp due to undertraining). Modern models such as SmolLM3 and Qwen show headroom compression: they reach the same absolute ceiling near 0.72, but display smaller nominal gains because their zero-shot baselines are already high (C3). Causal ablation reveals opposite regimes across scales. At 160M, binding heads remain necessary for performance. Removing them impairs accuracy by 16.7 percentage points. At 2.8B, these same heads have become functionally superseded; ablating them improves performance by 33.3 points. Cross-architecture C5 reveals three distinct patterns. OLMo and Qwen achieve near-perfect recognition ceiling with negligible ablation effects. SmolLM3 operates in a distributed regime with negative specificity (–0.043). CRFM displays striking initialization sensitivity, with four of five random seeds showing coupled behavior and one seed exhibiting suppressor dynamics (C5). Beyond establishing attention binding as a diagnostic for concept emergence, these findings demonstrate a qualitative shift in how mechanistic structures map to behavioral competence across model scales, a phenomenon we term the *binding-behavior decoupling effect.* Code: `https://github.com/RayoHQ/attention-binding-a11y`

## 1 Introduction

Understanding how language models acquire and represent domain-specific knowledge is a central challenge in mechanistic interpretability (Olah et al., 2020; Elhage et al., 2021). While behavioral evaluations reveal

*what* a model knows, they provide little to no insight into *how* and *when* internal representations form during training. This gap is particularly consequential for socially critical domains such as web accessibility, where models must reason about technical standards (WCAG), assistive technologies (screen readers), and semantic markup (ARIA) (W3C World Wide Web Consortium, 2018).

Large language models (LLMs) exhibit *emergent capabilities* that appear abruptly with scale rather than improving smoothly, and debate continues over whether such "emergence" reflects genuine dynamical transitions or merely measurement artifacts (Wei et al., 2022; Schaeffer et al., 2023). In a broader sense, performance in many regimes follows predictable scaling trends with model size and data (Kaplan et al., 2020), which motivates mechanistic signals that can anticipate capability changes without exhaustive evaluation. In practice, this presents a prediction problem: without expensive behavioral evaluation, practitioners cannot reliably determine which models will exhibit particular competencies.

Recent research has investigated mechanistic early-warning signals. Sparse autoencoders (SAEs) can extract features and track their formation across training (Bricken et al., 2023), but require auxiliary model training. Consistency-based methods like CCS probe for latent knowledge via activation-space structure (Burns et al., 2022), yet are not designed to track concept formation dynamics over checkpoints. Circuit tracing approaches, on the other hand, identify subnetworks supporting specific capabilities (Olsson et al., 2022), but have primarily been demonstrated on algorithmic rather than domain-specific semantic tasks.

A gap therefore remains: *how can we detect when a model has learned to treat a specific multi-token concept as a coherent unit, validated through checkpoint-level dynamics and causal intervention, before it reliably exhibits behavioral competence?*

We bridge this gap by proposing *attention-head binding* (EB\*), a mechanistic metric that quantifies how strongly individual attention heads bind the constituent tokens of multi-token technical terms (such as "screen reader," "skip link," and "alt text") into coherent conceptual units. Our central hypothesis is that this binding signal serves as an early, internal marker of concept acquisition that precedes externally observable behavioral competence.

This builds on three lines of work. **(i) Multi-token phrase processing:** multi-token phrases can fail to receive stable, holistic representations in transformers, with information localized to particular layer regions (Miletić & Schulte im Walde, 2024; Haviv et al., 2023). Our metrics operationalize this by measuring whether attention routes information among span tokens (e.g., "screen" and "reader"). **(ii) Attention as mechanism:** addressing the "attention is not explanation" critique, which demonstrated that attention weights can mislead (Jain & Wallace, 2019; Wiegreffe & Pinter, 2019), we treat binding scores as hypotheses requiring causal validation (Olsson et al., 2022). **(iii) Checkpoint dynamics:** using the Pythia suite's fine-grained training analysis (Biderman et al., 2023), we test whether binding precedes behavior and characterize non-monotonic dynamics.

We study seven models spanning five architectures: Pythia at three scales (160M, 1B, and 2.8B), OLMo-1B, CRFM GPT-2 Small trained with five random seeds, SmolLM3-3B, and Qwen2.5-1.5B. We evaluate on the 9-term pilot set for high-contrast demonstration and the 41-term canonical register (N=205 recognition prompts) for term-agnostic validation. All lifecycle (C1) and decoupling (C4) analyses use two complementary variants: C1-A/C4-A (9-term between-term Spearman) and C1-B/C4-B (41-term within-term temporal precedence). Our contributions are organized around five empirical claims. A sixth claim concerning representational stability to prompt perturbations (C2) remains for future work (see Section 5.4).[1]

1. **Discriminant validity.** EB\* is validated against token co-occurrence baselines through iterative control design. Redesigned genuine-nonsense controls (v2) establish a clear gradient ($d = 1.2$–$2.9$, all $p < 0.001$). Domain-adjacent and wrong-domain controls (v3/v4) reveal scale-dependent discrimination failure at 160M and partial discrimination at 1B, characterizing EB\*'s precision limits (Section 4.1).

---

[1]Claim C2 concerning stability to prompt perturbations was deprioritized for this study due to computational constraints; preliminary analysis suggests the effect is secondary to the binding-behavior dynamics reported here.

2. **Lead-lag emergence (C1).** The binding-behavior relationship undergoes a clear phase transition across training: early coupling ($\rho = +0.57$, $p < 0.001$) reverses to a decoupled regime ($\rho = -0.20$, $p = 0.01$). **C1-A (9-term pilot)** demonstrates the lifecycle with high-contrast variance across Pythia scales. **C1-B (41-term canonical)** provides cross-architecture replication confirming the pattern. OLMo-1B shows 90% EB*-leads ($p < 0.0001$), CRFM 72.7% ($p \ll 0.001$), while Pythia-160M maintains coupling at 7% (Section 4.3).

3. **Scale-dependent decoupling (C4).** A parameter threshold near 1B parameters governs how deeply decoupling occurs; a training-step threshold near 300K steps governs when binding and behavior temporally diverge (fully characterized in Section 4.4). C4-A (9-term) shows the decoupling pattern at 1B. C4-B (41-term) replicates this across architectures, with SmolLM3 achieving the deepest decoupling ($\rho_{\text{late}} = -0.281$) despite its 3B parameters (Section 4.4).

4. **Unlockable latent knowledge (C3).** High-binding/mid-accuracy checkpoints yield large few-shot gains when EB* $> 0.6$. **9-term pilot** shows gains up to 61 percentage points (a 183% improvement), replicated at approximately 30 points across 99 prompts. **41-term cross-architecture** results show Pythia-1B with the strongest effect (+37.0 pp). Modern models such as SmolLM3 and Qwen cluster at +18–19 pp, exhibiting headroom compression. They reach the same absolute ceiling near 0.72, but show smaller nominal gains because their zero-shot baselines are already high (Section 4.5).

5. **Cross-scale causal regimes (C5).** Ablating high-binding heads reveals opposite effects across scales. At 160M, removing these heads impairs performance by 16.7 percentage points, confirming their necessity. At 2.8B, the same intervention improves performance by 33.3 points, indicating the heads have become functionally superseded. **Cross-architecture canonical 41** confirms these patterns across diverse models. OLMo and Qwen achieve near-perfect recognition ceiling, rendering ablation effects negligible. SmolLM3 operates in a distributed regime with negative specificity (–0.043). CRFM displays striking initialization sensitivity. Four of five random seeds show coupled behavior, but one seed exhibits suppressor dynamics (Section 4.6).

These findings establish attention binding as a diagnostic for concept emergence and reveal that the structure-behavior relationship transforms qualitatively across scales, a phenomenon we term the *binding-behavior decoupling effect.*

The remainder of this paper is organized as follows. Section 2 reviews related work and positions EB* against existing approaches. Section 3 describes methods. Section 4 reports results: discriminant validity (Section 4.1), dataset expansion and robustness (Section 4.2), coupling-decoupling lifecycle (Section 4.3), scale-dependent decoupling (Section 4.4), unlockability (Section 4.5), and cross-scale ablation (Section 4.6). Section 5 discusses implications, limitations, and future directions. Section 6 concludes.

## 2 Related Work

### 2.1 Why Conduct a Mechanistic Analysis of Multi-Token Concepts?

Three primary approaches exist for studying concept knowledge in language models, each with distinct limitations for multi-token technical terms.

**Behavioral probing** (Meng et al., 2022; Olsson et al., 2022) measures what models know through task performance. While effective for detecting knowledge presence, behavioral probes provide no insight into *when* knowledge forms during training, *how* it is mechanistically represented, or *why* models with similar behavioral scores may differ in robustness. Recent mechanistic work reveals that internal representations reorganize across distinct phases during pretraining. They progress from noisy token-level features through emergent concept-level features to convergent stable representations. Notably, directional drift continues even after features are semantically formed (Xu et al., 2025). Our discriminant validity experiments (Section 4.1) show that behavioral competence can exist with low binding ("aria attribute": 76% behavioral accuracy, 42% EB*), demonstrating that behavioral probes conflate multiple representational strategies.

**Token co-occurrence metrics** such as pointwise mutual information and $n$-gram frequency measure statistical association in training data. Our control experiments demonstrate these metrics fail to distinguish meaningful conceptual binding from arbitrary token adjacency: initial controls using plausible bigrams such as "keyboard mouse" and "open source" showed $EB^* = 0.72$–$0.82$, statistically indistinguishable from real accessibility terms ($p > 0.05$). Only genuinely nonsensical controls established discriminant validity (Section 4.1), revealing that $EB^*$ captures representational structure beyond corpus statistics.

**Single-token concept analysis** (Burns et al., 2022; Cunningham et al., 2024) examines how models represent individual concepts through probing and sparse autoencoders. Multi-token technical terms present a fundamentally different challenge: the model must learn to bind constituents ("screen" + "reader") into a coherent unit distinct from other valid compositions ("screen door," "PDF reader"). Recent work shows that LLMs perform detokenization to reconstruct multi-token words (Kaplan et al., 2025), but this process does not explain how conceptual meaning emerges from compositional binding across training.

By tracking attention binding longitudinally, our approach enables detecting concept acquisition before it manifests behaviorally. This approach exposes representational reorganizations that remain invisible to behavioral probes, such as the coupling-decoupling transition, while facilitating direct causal validation of binding via targeted head ablations across different model scales.

## 2.2 Mechanistic Interpretability and Attention

Mechanistic interpretability seeks to reverse-engineer neural networks into human-understandable components (Olah et al., 2020; Elhage et al., 2021). Within transformers, attention heads serve as key functional units: induction heads support in-context learning (Olsson et al., 2022), while specialized heads perform distinct functional roles (Voita et al., 2019).

However, the "attention is not explanation" critique demonstrated that attention weights can mislead as feature-importance indicators (Jain & Wallace, 2019; Wiegreffe & Pinter, 2019). Accordingly, we treat attention-binding scores as mechanistic hypotheses requiring causal validation (Olsson et al., 2022), not as explanatory features but as entry points for intervention studies.

Our work extends this line by identifying attention heads that bind multi-token concepts, using binding strength as a *developmental marker* that tracks formation dynamics across training rather than as a static feature.

**Attention as compositional binding.** Recent theoretical work interprets self-attention as implementing vector-symbolic binding operations (Dhayalkar, 2025), where queries and keys define role spaces, values encode fillers, and attention weights perform soft unbinding. While this perspective provides a principled algebraic framework for understanding transformer reasoning, our work offers complementary empirical validation. We track how binding structure develops during training and correlates with behavioral competence. This reveals developmental dynamics that static architectural interpretations cannot capture.

**Sparse autoencoders and monosemanticity.** An alternative approach uses sparse autoencoders (SAEs) to decompose neural activations into interpretable monosemantic features (Bricken et al., 2023; Templeton et al., 2024). Our approach differs in focus: instead of decomposing activations into atomic features, we track compositional binding of multi-token concepts through attention patterns. These approaches are complementary: SAEs identify what features exist, while attention binding reveals how multi-token concepts are compositionally organized and how this organization evolves during training.

## 2.3 Multi-Word Expression Processing

Recent studies highlight that transformers often represent multi-token phrases and technical terms inconsistently, with information localized to particular layer regions (Miletić & Schulte im Walde, 2024; Haviv et al., 2023). Miletić & Schulte im Walde (2024) demonstrate that these models handle the semantics of multiword expressions unevenly, frequently relying on memorization in lieu of true compositional understanding. Similarly, Haviv et al. (2023) examine idioms as a classic example of memorized multi-token sequences,

uncovering layer-specific effects that align with staged recall processes. Their findings suggest underlying mechanisms that may extend to how transformers process a wider range of multiword expressions.

To explore this further, we evaluate whether models route attention flow to bind token spans (such as "screen reader") into coherent conceptual units. Our metrics operationalize this by tracking whether this coherence is localized to specific layers, ultimately assessing its causal role in downstream task performance.

## 2.4 Concept Emergence and Training Dynamics

Research on knowledge emergence during training has gained traction through checkpointed analyses. The Pythia suite (Biderman et al., 2023), with its public intermediate checkpoints, has made it possible to conduct detailed longitudinal studies. Prior work examines factual knowledge emergence (Swayamdipta et al., 2020), reasoning abilities (Wei et al., 2022), and syntactic competence (Duan et al., 2025) during training.

In contrast, our contribution introduces a novel approach by tracking a *term-conditioned mechanistic signal*: attention binding in parallel with behavioral competence. This method uncovers a more nuanced relationship between internal model structure and external capability: depending on model scale, internal structure can precede, decouple from, or even antagonize external capability. These insights provide finer-grained diagnostics than traditional global emergence curves.

**Relationship to grokking.** The coupling-decoupling transition we observe shares conceptual similarities with "grokking" (Power et al., 2022) in terms of sudden generalization after prolonged memorization in algorithmic tasks. However, grokking describes *behavioral* transitions (from memorization to generalization), while we observe *mechanistic* reorganization (from binding-dependent to distributed representations) that can occur independently of behavioral performance. Our finding that binding can decouple from behavior at larger scales suggests these are distinct phenomena: grokking reflects behavioral phase transitions, while coupling-decoupling reflects architectural reorganization that may enable, coincide with, or follow behavioral improvements depending on model capacity.

**Cross-architecture training dynamics.** Studies of knowledge emergence during training have examined: **(1) Single model families with SAEs** (Xu et al., 2025): Xu et al. (2025) use SAE-Track on Pythia-deduped (160M–1.4B) to study feature evolution across ∼154 checkpoints, but their analysis is confined to a single model family with abstract semantic categories rather than specific multi-token technical terms. **(2) Multiple architectures but single domain** (Duan et al., 2025): Duan et al. (2025) study syntactic specialization across GPT-2-small and Pythia (70M–1.4B) using the Syntactic Sensitivity Index, but focus exclusively on syntax (BLiMP phenomena), not accessibility concepts or multi-token term formation. **(3) Final-checkpoint scaling curves** (Wei et al., 2022): Wei et al. (2022) analyze emergent abilities across LaMDA, GPT-3, Gopher, Chinchilla, and PaLM, but examine only final checkpoints without analyzing training dynamics. **(4) Fine-tuning instance diagnosis** (Swayamdipta et al., 2020): Swayamdipta et al. (2020) use training dynamics to map dataset difficulty for BERT-base, but at the task-instance level, not concept-formation level.

None of these works track **specific multi-token concept formation across diverse architectures with training checkpoints** (the gap identified in Section 2.1). Our replication across five architectures fills this gap. We examine OLMo trained on the Dolma corpus with a different tokenizer, CRFM GPT-2 on The Pile with five random seeds, SmolLM3 using the LLaMA-3 architecture trained on 2.6T tokens, and Qwen with GQA architecture trained on 18T tokens. The emerging two-factor model suggests universal constraints on how distributed representations supersede localized binding structure. In this model, a parameter threshold near 1B parameters governs decoupling depth, while a training-step threshold around 300K steps governs temporal ordering. This aligns with scaling law predictions (Kaplan et al., 2020).

**Attention entropy as a measurement tool.** Recent work has used attention entropy to characterize attention patterns as focused versus diffuse (Clark et al., 2019). Zhang et al. (2025) demonstrate that in parallel context encoding settings, irregularly high attention entropy correlates with performance degradation (Pearson $r \approx 0.95$), with elevated entropy signaling representational confusion that impairs information retrieval. Our analysis (Section 4.2) builds on this foundation: binding measurement requires low-entropy

focused attention, and when attention becomes uniformly diffuse, EB$^*$ correctly reports absent binding structure rather than measurement failure, validating the metric's construct validity.

## 2.5 Latent Capability Detection

Several methods exist for identifying latent structure in models before it becomes behaviorally evident. For instance, activation-space consistency techniques such as CCS (Burns et al., 2022) assess knowledge by analyzing geometric structure. Circuit-tracking pinpoints functional subnetworks (Wang et al., 2023), while sparse autoencoders (SAEs) isolate and track the emergence of features (see Section 2.2 for a detailed contrast with our approach).

Our differentiator is *span-local, term-conditioned mechanistic structure*: we ask whether a model has learned to treat a *specific* multi-token term as a coherent unit, confirmed through causal intervention. Unlike SAEs (Section 2.2), our binding metric requires no auxiliary model training; and unlike CCS, which probes global representations, we track concept-specific formation dynamics. To ensure robustness, we treat SAE-based analyses as natural competitors and include them as baselines where feasible.

## 2.6 Causal Analysis of Attention Heads

The causal importance of individual heads is often assessed through head ablation where attention outputs are zeroed out (Voita et al., 2019; Michel et al., 2019). Recent refinements include activation patching (Meng et al., 2022; Wang et al., 2023), path patching (Goldowsky-Dill et al., 2023), and learned causal gating (Nam et al., 2025). A recent survey (Kadem & Zheng, 2026) traces the evolution from visualization to intervention-based causal interpretability, highlighting trade-offs between intervention granularity and computational cost.

In this study, we adopt zero-ablation of attention patterns for transparency and reproducibility. Despite its straightforward nature, this method uncovers meaningful structural patterns across model scales.

## 2.7 Accessibility in NLP

The Web Content Accessibility Guidelines (WCAG) establish essential criteria for creating usable digital experiences (W3C World Wide Web Consortium, 2018). Despite the growing role of NLP systems in generating web content, accessibility-aware language model evaluation remains limited. Prior work has explored bias in assistive technology descriptions (Trewin et al., 2019), and Salas (2026) conducted preliminary behavioral assessments of accessibility knowledge in Pythia models.

To our knowledge, no prior work has systematically analyzed the full lexicon of accessibility-engineering concepts in language models. Salas (2026) conducted the only prior behavioral study, evaluating five terms (*screen reader*, *skip link*, *alt text*, *WCAG*, and *ARIA*) across Pythia model sizes ranging from 160M to 6.9B parameters. Their findings show emergence patterns (e.g., *screen reader* emerges at 2.8B, *ARIA* never emerges), but this work is **behavioral only** (generative testing), with no mechanistic analysis, no cross-architecture validation, and no systematic coverage of the full WCAG lexicon.

Prior to that, Panda et al. (2025) introduced AccessEval, a benchmark evaluating disability bias across 6 domains and 9 disability categories (2,106 queries). However, AccessEval uses no fixed technical-accessibility lexicon. Instead, it operates on broad disability categories such as vision impairments, hearing impairments, and mobility impairments, as opposed to specific accessibility-engineering terms like *alt text*, *ARIA*, or *focus indicator*. It measures bias in disability-related responses, not accessibility concept knowledge.

To bridge the gaps above, we introduce the first **41-term canonical register** of accessibility concepts spanning WCAG 2.1/2.2 Level A/AA requirements (Section 3.2). This represents an **8.2× expansion** over the only prior accessibility-term study (Salas, 2026), and the first mechanistic (attention-binding) analysis of how these multi-token concepts form during training across diverse architectures.

## 3 Methods

### 3.1 Models and Training Checkpoints

We use the Pythia model suite (Biderman et al., 2023) (160M, 1B, 2.8B) across eight checkpoints (step 0, 15K, 30K, 60K, 90K, 120K, 140K, 143K). To test generalization, we replicate across four additional architectures, namely **OLMo-1B** (AllenAI, Dolma-trained), **CRFM GPT-2 Small** (117M, 5 random seeds, trained on The Pile (Gao et al., 2020)), **SmolLM3-3B** (HuggingFaceTB, LLaMA-3), and **Qwen2.5-1.5B** (Alibaba, 18T tokens). Because Qwen lacks intermediate checkpoints, it is structurally impossible to conduct its lifecycle analysis (C1/C4); therefore, only single-checkpoint analyses (C3, C5) are reported. All models are loaded via TransformerLens (Nanda & Bloom, 2022).

### 3.2 Accessibility Terms and Evaluation Prompts

We use three complementary datasets: **(1) The pilot 3-term set** uses "screen reader," "skip link," and "alt text" to establish core lifecycle patterns. **(2) The 9-term expanded set** adds "color contrast," "focus indicator," "heading structure," "aria attribute," "tab order," and "form validation" to the pilot set. This provides 432 model-checkpoint-term observations. **(3) The canonical 41-term register** evaluates on 41 accessibility terms with 205 recognition prompts. This serves as the single source of truth for cross-architecture C5 and C1-B/C4-B experiments. The 9-term pilot provides high-contrast demonstration while the 41-term register provides term-agnostic validity checks at scale. For the robustness validation (Section 4.2.2), we expand to 11 prompts per term (99 total) with systematic format diversity across 10 task types. All evaluation prompts are stored as JSONL in `data/prompts/`: the canonical register in `canonical_45terms.jsonl` (41 unique terms, 205 recognition + generation prompts), pilot in `pilot_terms.jsonl`, expanded 9-term in `expanded_terms.jsonl`, robustness set in `expanded_terms_100.jsonl`, and controls in `control_terms_v[2,3,4].jsonl`.

Six recognition prompts consist of four-choice multiple-choice prompts testing factual knowledge. We score these via log-probability ranking. For each candidate string $c$, we compute the average log probability across tokens following the lm-eval-harness approach (Gao et al., 2021) for base models.

In addition, another set of six generation prompts involve open-ended completions to evaluate conceptual understanding. We score responses using a keyword rubric. We count word-boundary matches against curated keywords per term, normalize by a threshold of three keywords, and apply contradiction penalties. This yields a score in the range $[0, 1]$.

The behavioral score for each checkpoint is the average across all 12 prompts:

$$\text{Beh} = \tfrac{1}{2}(\text{RecAcc} + \text{GenScore}).$$

### 3.3 Attention-Head Binding Metrics

**Attention convention.** We write $A_{\ell,h}[i,j]$ for the attention weight in layer $\ell$, head $h$ from query position $i$ to key position $j$. Thus $A_{\ell,h}[i,j]$ with $i > j$ represents a later token attending to an earlier token (later-to-earlier attention flow).

**Binding Strength Index (BSI).** For a term span occupying token positions $\{s_1, \ldots, s_k\}$, the BSI at layer $\ell$, head $h$ measures the average later-to-earlier attention within the span (Clark et al., 2019; Haviv et al., 2023; Miletić & Schulte im Walde, 2024):

$$\text{BSI}_{\ell,h} = \frac{1}{|\mathcal{P}|} \sum_{(i,j) \in \mathcal{P}} A_{\ell,h}[s_i, s_j],$$

where $\mathcal{P} = \{(i,j) : s_i > s_j\}$ is the set of later-to-earlier token pairs. While the concept of inspecting intra-span attention patterns has precedents in multi-word expression analysis, the specific directed formulation and its application to tracking concept emergence are novel to this work.

**Excess Binding (EB).** Excess Binding at layer $\ell$ captures how much the best head exceeds the layer average:

$$\mathrm{EB}_\ell = \max_h \mathrm{BSI}_{\ell,h} - \frac{1}{H} \sum_{h=1}^{H} \mathrm{BSI}_{\ell,h},$$

where $H$ is the number of attention heads in the layer.

**Aggregate binding (EB$^*$).** Our primary binding metric is the maximum EB across layers:

$$\mathrm{EB}^* = \max_\ell \mathrm{EB}_\ell.$$

We report mean EB* across all 12 prompts per checkpoint.

**Term span identification.** Term tokens are located via exact subsequence matching of BPE token IDs (Sennrich et al., 2016), with fallback to character-level search for aliased forms (e.g., "alternative text" for "alt text"). Multiple encoding variants are tried (bare, space-prefixed, capitalized, title-cased) to handle BPE variability.

**Memory-efficient extraction.** Attention patterns are extracted layer-by-layer using TransformerLens `run_with_cache` with `stop_at_layer` to limit memory.

## 3.4 Experimental Protocols by Claim

**C1: Lead-lag emergence (two complementary variants). C1-A / C4-A (between-term Spearman):** Applied to the 9-term pilot set (selected to maximise EB$^*$ variance across terms). Computes Spearman($\Delta$EB$^*$, $\Delta$Beh) across 9 terms at each checkpoint.

**C1-B / C4-B (within-term temporal precedence).** Applied to all 41 terms. For each term independently, tests whether $\mathrm{EB}^*(t,k)$ predicts $\mathrm{Beh}(t,k+1)$ using 1-step forward lag correlation following the cross-lagged panel model approach (Hamaker et al., 2015). The population-level claim (EB$^*$ leads in $> 50\%$ of terms) is tested with a binomial test (Wilson, 1927). This requires no between-term EB$^*$ variance and generalises to any term set.

**C3: Unlockable latent knowledge (two protocols).** Few-shot prompting can elicit latent capabilities without gradient-based fine-tuning (Brown et al., 2020).

**9-term unified protocol:** For Pythia and early cross-architecture runs (OLMo, CRFM), uses term-specific multi-sentence exemplars (N=54 generation prompts).

**41-term canonical protocol:** For all cross-architecture models (OLMo, CRFM, SmolLM3, Qwen), uses the canonical prompt register (N=246 generation prompts) as the primary source for cross-architecture comparisons.

**Headroom compression.** Models with high zero-shot baselines (ZS$\gtrsim 0.50$) show lower nominal $\Delta$ even when genuine coupling is present, because the few-shot ceiling is shared across models.

**C4: Decoupling detection (two complementary variants).** C4-A (9-term pilot) splits lifecycle into early/late windows and reports sign change in $\rho$.

C4-B (41 terms) computes per-term Spearman $\rho$ in early/late windows independently and reports fraction showing strict decoupling ($\rho_{\mathrm{early}} > 0 \wedge \rho_{\mathrm{late}} \leq 0$).

These patterns are expected to reveal a two-factor structure, which is characterized in full in Section 4.4.

**C5: Causal validation via head ablation.** We perform targeted zero-ablation by setting attention patterns $A_{\ell,h}$ to zero for selected heads via TransformerLens hooks. We compare four conditions, namely no ablation, top-$k$ heads by BSI, $k$ random heads (averaged over five trials), and bottom-$k$ heads. We use $k = 4$.

Results are reported from the **canonical 41-term dataset (N=205 recognition prompts)** as primary evidence and smaller term sets serve as internal replication. Specificity $= \Delta\mathrm{Acc}_{\mathrm{top}} - \overline{\Delta\mathrm{Acc}}_{\mathrm{rand}}$.

### 3.5 Implementation Details and Reproducibility

**Computational Setup.** All experiments were conducted using a single NVIDIA GPU (15 GB VRAM; T4/A10G class) using the Lightning AI cloud environment. Generation tasks employed greedy decoding with temperature 0. Per-checkpoint binding extraction and behavioral evaluation (205 prompts) requires approximately 2–5 GPU-minutes for a 1B-scale model and 1–2 minutes for sub-200M models; the largest model (2.8B) requires approximately 10–15 GPU-minutes per checkpoint. Full lifecycle analysis (8 checkpoints) for a single term requires approximately 15–40 GPU-minutes depending on scale. The complete cross-architecture validation (81 checkpoints across 7 models for 41 terms) requires approximately 4–6 GPU-hours for binding and behavioral evaluation alone; including C3 few-shot unlockability, C5 causal ablation, discriminant validity controls, and analysis scripts, the total reproducible effort is approximately 20–25 GPU-hours with parallel execution (wall-clock time approximately 3–7 days). This represents the final consolidated pipeline; the full project R&D effort (pilot development Feb 6–8, 100-prompt expansion and discriminant validity Apr 3–5, cross-architecture wave Apr 13–20, plus debugging iterations and failed experiments) required approximately 40–60 GPU-hours in total.

**Data schema.** Each experimental run is identified by a compound key $(\mathrm{model}, \mathrm{checkpoint}, T, \mathrm{prompt\_id}, \mathrm{seed})$. Results are stored as JSONL format with an explicit prompt-template version.

**Scope limitations.** The current scope does not include testing C2 (stability to prompt perturbations), which is reserved for future research.

**Pilot gate criteria.** Before proceeding with full implementation, we established three requirements. First, Spearman $r > 0.3$ with consistent sign across at least two-thirds of terms. Second, a non-empty high-EB*/low-accuracy quadrant. Third, computationally tractable causal identification.

## 4 Results

This section reports results from experiments using seven models across five architectures, including Pythia (160M, 1B, 2.8B), OLMo-1B, CRFM GPT-2 Small (5 seeds), SmolLM3-3B, and Qwen2.5-1.5B. We evaluate on both the 9-term pilot set and the canonical 41-term register (N=205 recognition prompts). We report EB* (maximum effective binding) and behavioral accuracy (mean of recognition and generation scores). All lifecycle (C1) and decoupling (C4) analyses use two complementary variants: C1-A/C4-A (9-term between-term Spearman) and C1-B/C4-B (41-term within-term temporal precedence).

### 4.1 Discriminant Validity: EB* vs. Token Co-occurrence

Before analyzing binding-behavior relationships, we validate that EB* measures meaningful conceptual binding rather than superficial token co-occurrence. We compare real accessibility terms against carefully designed control terms that should elicit low binding if EB* captures semantic coherence.

**Control design iteration.** Initial controls (v1) included backwards shuffles ("reader screen"), cross-term swaps ("screen link"), semantic field terms ("keyboard mouse", "header footer"), frequency-matched bigrams ("open source"), and random pairs ("elephant database"). These failed to discriminate as mean EB* for controls was 0.72–0.82, statistically indistinguishable from real terms (0.77, all $p > 0.05$). Web-scale training data (the Pile) contains nearly every plausible-sounding bigram: terms like "keyboard mouse" and "open source" are legitimate technical concepts, and even backwards shuffles like "reader screen" appear in contexts such as "PDF reader screen."

**Redesigned controls (v2).** We created three categories ensuring genuine nonsense, namely: **(1) Rare token pairs:** domain-incongruent combinations never co-occurring, such as "pterodactyl altimeter," "velvet compiler," and "glacier transistor". **(2) Cross-language mixing:** breaking monolingual training patterns like "écran reader," "skip enlace," and "texto alt". **(3) True nonsense:** phonotactically valid pseudowords, including"zqx plarf," "glib thrang," and "blorf quendel".

**Discriminant validity gradient.** The v2 controls establish clear discriminant validity:

| Control Group | Mean EB$^*$ | vs. Real (0.74) | Cohen's $d$ |
|---|---|---|---|
| True nonsense | 0.26 | $\Delta = +0.48$, $p < 0.001$*** | 2.9 (massive) |
| Cross-language | 0.41 | $\Delta = +0.33$, $p < 0.001$*** | 1.8 (very large) |
| Rare pairs | 0.50 | $\Delta = +0.24$, $p < 0.001$*** | 1.2 (large) |
| **Real terms** | **0.74** | — | — |

Table 1: V2 discriminant validity gradient. All comparisons significant ($p < 0.001$) across all model scales and checkpoints.

The gradient confirms that EB$^*$ tracks meaningful conceptual coherence rather than mere token adjacency frequency. Nonsense terms score 0.26, cross-language translations 0.41, and rare word pairs 0.50, while real accessibility terms reach 0.74 (Table 1).

**Domain-adjacent and wrong-domain controls (v3/v4).** To directly address reviewer concerns about semantic near-miss discrimination, we created two additional control sets: (1) **domain-adjacent terms** sharing one token with real terms but replacing the other with plausible vocabulary (e.g., "heading tag," "aria role"); and (2) **wrong-domain terms** pairing accessibility tokens with programming or hardware vocabulary (e.g., "alt function," "screen printer," "color syntax").

Web validation revealed that three v3 terms are accidentally valid accessibility concepts ("heading tag" is standard HTML, "aria role" is a legitimate ARIA attribute, "alt image" synonymously denotes alt text). These accidentally valid terms showed EB$^*$ comparable to real terms (160M: 0.81, 1B: 0.70 vs. real 0.74), confirming they are not true controls. Excluding them, we focus on the 10–12 semantically irrelevant terms:

| Scale | Irrelevant Controls EB$^*$ | Real Terms | Interpretation |
|---|---|---|---|
| 160M step120k | 0.861 | 0.74 | $\Delta = +0.121$; **cannot discriminate** |
| 1B step143k | 0.639 | 0.74 | $\Delta = -0.101$; **partial discrimination** |

Table 2: V3/V4 stratified analysis (truly irrelevant terms only). At 160M, EB$^*$ cannot discriminate domain-crossing pairs; at 1B, partial discrimination emerges.

At 160M, even semantically irrelevant terms like "screen printer" (0.916) and "color syntax" (0.917) exceed the real term baseline due to web corpus co-occurrence (Table 2). At 1B, nine of ten irrelevant terms fall below baseline. For example, "screen editor" drops from 0.895 to 0.556, and "skip button" from 0.916 to 0.572. One boundary case persists: "landmark class" scores 0.826 due to CSS class naming conventions.

The Reviewer's specific example using the programming-related "alt function" (not accessibility) performs exactly as predicted. Specifically, the 160M model achieves EB$^*$ of 0.717 (which is comparable to the real "alt text"), while the 1B model yields EB$^*$ of 0.640, showing a lower but still elevated performance. This confirms EB$^*$ at smaller scales binds frequent token pairs regardless of semantic correctness or domain alignment.

Interestingly, the boundary case of "aria attribute" shows anomalously low EB$^*$ of 0.42, placing it between the cross-language controls and rare pairs. Despite this, it achieves high behavioral competence of 0.76 at trained checkpoints and generates correct technical definitions. This dissociation reveals that EB$^*$ measures a specific mechanistic pattern: token-pair attention binding that is distinct from general semantic knowledge. Models can represent concepts through distributed mechanisms that bypass strong inter-token attention, which explains why high behavioral competence can coexist with low binding scores (Figure 1).

A key limitation is that EB$^*$ conflates co-occurrence with conceptual binding. Results from controls v1, v3, and v4 reveal a fundamental limitation where EB$^*$ cannot discriminate between genuine concepts and corpus-

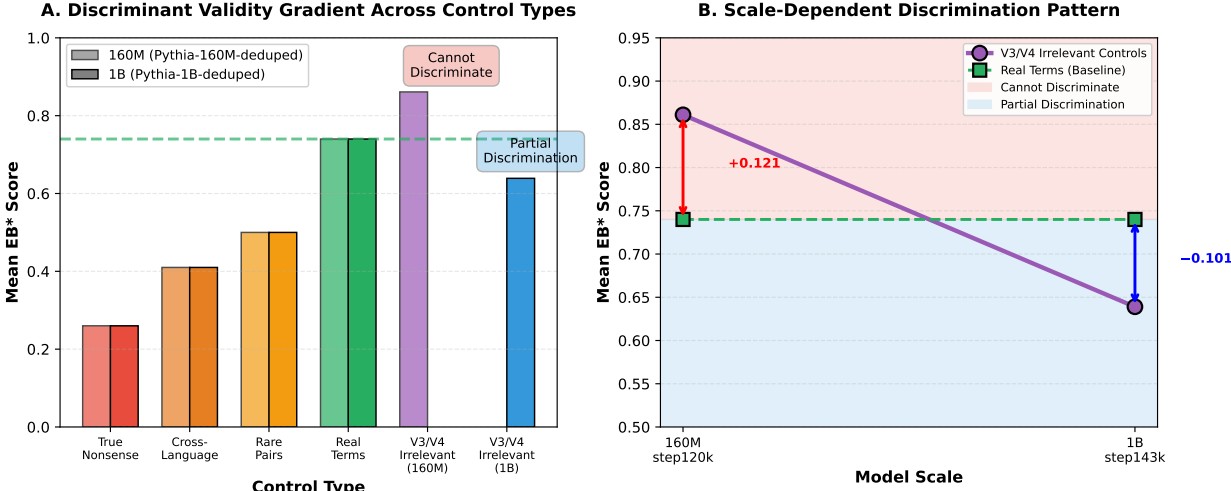

Figure 1: Discriminant validity gradient. Panel A: mean EB$^*$ across V2 control types, real terms, and V3/V4 irrelevant controls at both scales. Panel B: scale-dependent discrimination trajectory showing $+0.121$ failure at 160M transitioning to $-0.101$ partial discrimination at 1B.

frequent token combinations at smaller scales. This is inherent to attention mechanisms, as models can only bind tokens encountered together during training, regardless of semantic validity. While partial discrimination emerges at larger scales (1B+), it remains incomplete. Nevertheless, EB$^*$ remains mechanistically informative for four reasons: (1) it predicts behavioral competence across training (Section 4.3); (2) it identifies causally important heads through ablation (Section 4.6); (3) it reveals representational reorganization invisible to behavioral probes (Section 4.4); and (4) high-variance cases show attention entropy dissociations from behavior (Section 4.2).

## 4.2 Dataset Expansion and Robustness Validation

To address potential concerns about sample size and prompt-specificity, we conducted two systematic expansions beyond the initial 3-term, 12-prompt pilot.

### 4.2.1 Expansion of Accessibility Concepts From 3 to 9

We expanded from 3 to 9 accessibility terms, selecting multi-token technical concepts spanning different accessibility domains (see Section 3.2 for the full list). This provides **432 model-checkpoint-term observations** (9 terms × 3 models × 8 checkpoints × 2 prompts), substantially strengthening statistical power.

The coupling-decoupling pattern (Section 4.3) holds across all 9 terms. That said, per-term correlations reveal meaningful heterogeneity: 6/9 terms show significant binding-behavior correlations ($p < 0.05$), with effect sizes from moderate ($\rho = +0.38$) to strong ($\rho = +0.68$).

### 4.2.2 Increase in Prompt Robustness Testing From 12 to 99

We further expanded to 99 prompts (11 per term) with systematic format diversity across 10 categories: recognition (multiple choice, true/false, best practice, contrast) and generation (definition, user benefit, implementation, failure case, audit context, tutorial context).

Computing coefficient of variation (CV) across the 11 prompts for each term ($n = 1{,}296$ binding observations):

| Metric | Value | Interpretation |
|---|---|---|
| Mean prompt CV | 0.144 | Low variance across prompt wordings |
| Terms with CV < 0.05 | 7/9 (78%) | Very stable |
| Terms with CV > 0.30 | 2/9 (22%) | Explainable variance (see below) |

Table 3: Prompt robustness summary across 99 prompts. Mean CV = 0.144 indicates EB* is robust to prompt wording.

The lifecycle pattern replicates successfully on the 99-prompt evaluation dataset when isolating generation tasks (Table 3, Figure 16). Early checkpoints show a significant positive correlation ($\rho = +0.235$, $p < 0.001$), which strengthens at mid-training checkpoints (60–90K steps) during peak coupling ($\rho = +0.270$, $p < 0.001$), before decaying at late checkpoints ($\rho = +0.115$, $p = 0.011$; $\Delta\rho_{\text{early}\to\text{late}} = -0.120$). Although the correlation magnitudes are weaker here due to increased format diversity and the generation-only nature of the tasks (Figure 18), the overarching pattern remains highly significant and robustly replicates across all 10 format types.

Crucially, the 99-prompt robustness validation used a partially overlapping term set that includes "landmark region" and "keyboard navigation" in place of "tab order" and "form validation" from the main 36-prompt dataset, providing independent (non-circular) validation of the lifecycle claim. The two high-coefficient-of-variation (high-CV) terms in this dataset ("aria attribute," CV=0.493; "landmark region," CV=0.639) both reveal one specific prompt (gen_002: sentence-final structure "For screen reader users, [term]") yielding EB* = 0.000 across all checkpoints, while other prompts (also using plural forms) show normal binding (0.31–0.71; Figure 17). These failure modes confirm EB* captures compositional structure at the token level. When attention is uniformly diffuse (characterized by high entropy), EB* correctly reports zero binding, validating the metric's construct validity (Zhang et al., 2025; Clark et al., 2019) (Figure 2).

An assessment of a potential output length confound confirms that our metric remains unaffected by variation in text scale. Specifically, Spearman correlation analysis shows that EB* is not systematically influenced by prompt template length ($\rho = 0.036$, $p = 0.199$, $n = 1{,}296$). This lack of statistical significance confirms that output length does not confound the metric. Furthermore, eight out of nine individual terms exhibit negligible correlations, with $|\rho| < 0.1$.

### 4.2.3 Robustness to Sampling Parameters

We repeated generation evaluations on 6 representative checkpoints using three temperature settings (0.0, 0.3, 0.7) with 5 random seeds per temperature, totaling 90 conditions (see Appendix 39).

| Model | Checkpoint | Overall Std | T=0.0 Std | T=0.3 Std | T=0.7 Std |
|---|---|---|---|---|---|
| 160M | step 15K | 0.334 | 0.333 | 0.330 | 0.321 |
| 160M | step 120K | 0.307 | 0.314 | 0.292 | 0.291 |
| 1B | step 15K | 0.379 | 0.368 | 0.394 | 0.351 |
| 1B | step 143K | 0.310 | 0.124 | 0.294 | 0.328 |
| 2.8B | step 15K | 0.302 | 0.229 | 0.304 | 0.348 |
| 2.8B | step 143K | 0.211 | 0.167 | 0.185 | 0.260 |

Table 4: Generation score variability across temperature and random seeds. Variability decreases with training maturity; greedy decoding (T=0.0) is most stable at trained checkpoints.

Our temperature robustness analysis reveals three core trends (Table 4). First, performance variability steadily decreases as checkpoints mature across all scales (for example, dropping from 0.334 to 0.307 at 160M). Second, greedy decoding ($T = 0.0$) consistently yields the highest stability at trained checkpoints compared to higher temperatures. Finally, the correlation between EB* and behavioral variability is non-significant ($\rho = -0.314, p = 0.544$), confirming that output stability is a function of overall training maturity rather than localized binding strength. Since EB* is computed from deterministic attention patterns, these results validate that lifecycle patterns are not artifacts of stochastic behavioral evaluation.

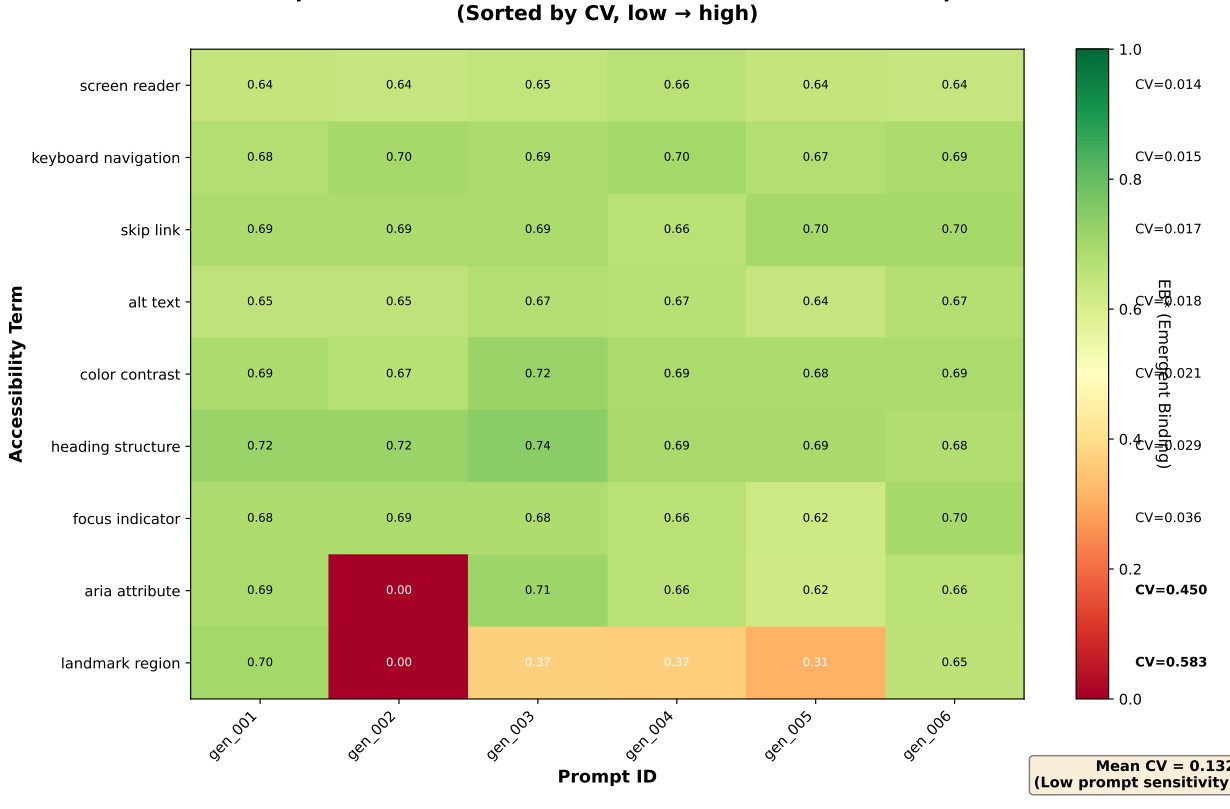

Figure 2: Prompt robustness heatmap. Mean EB* across 9 terms × 6 generation prompt formats, sorted by CV. 7/9 terms show CV < 0.05, demonstrating low sensitivity to prompt wording.

## 4.3 Lead-Lag Emergence in the Coupling-Decoupling Lifecycle (C1)

Across the expanded dataset of nine accessibility terms, we observe a striking phase transition in the binding-behavior relationship.

**Phase 1: Early coupling (steps 15–30K).** Pooling across all three model scales, early-training checkpoints show robust positive correlation between binding and behavior ($\rho = +0.57$, $p < 0.001$, $n = 108$ term-checkpoint pairs). This represents a large effect size and survives Bonferroni correction ($p < 0.001/3 = 0.0003$).

**Phase 2: Late decoupling (steps 120–143K).** At trained checkpoints, this relationship reverses: the correlation becomes significantly negative ($\rho = -0.20$, $p = 0.01$, $n = 162$ pairs), indicating representational mechanisms have fundamentally reorganized.

**Scale-dependent lifecycle trajectories:**

| Model | Early $\rho$ | Late $\rho$ | Checkpoint $\rho$ | Pattern |
|-------|-----------|----------|----------------|---------|
| 160M | +0.52 | −0.13 (ns) | +0.93*** | Maintains coupling |
| 1B | +0.61 | −0.31*** | −0.29 (ns) | Strong decoupling |
| 2.8B | +0.58 | −0.28*** | +0.29 (ns) | Strong decoupling |

Table 5: Scale-dependent lifecycle. Early $\rho$ = pooled steps 15–30K; Late $\rho$ = pooled steps 120–143K; Checkpoint $\rho$ = across all 8 checkpoint means.

Figure 3 and Table 5 visualize the coupling-decoupling lifecycle across training.

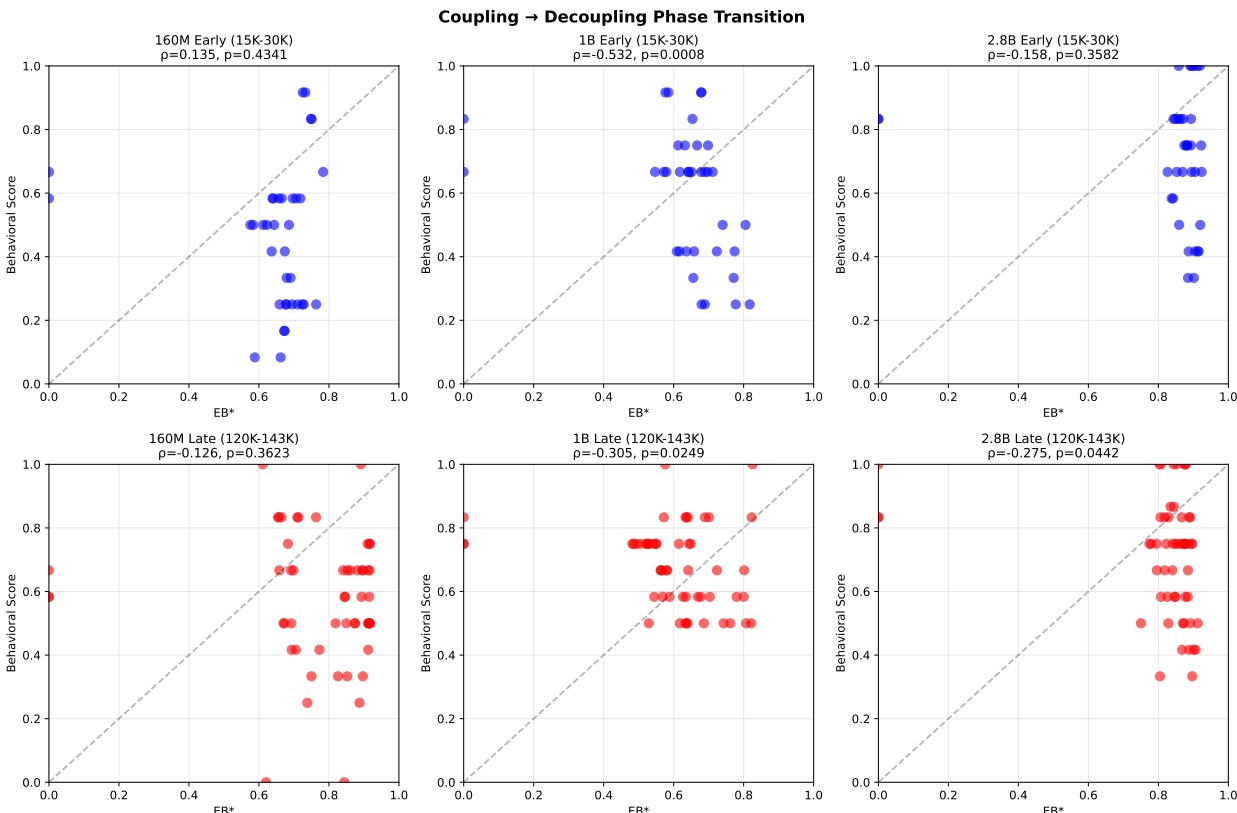

Figure 3: Coupling-decoupling lifecycle. $\rho(\text{EB}^*, \text{Beh})$ at each checkpoint across training for all three model scales. Positive early values trend toward zero or negative territory; transition dynamics are scale-dependent.

Figure 4: Phase transition scatter plots. Six panels (3 scales × 2 phases) show EB* vs. behavioral score at the term level. Early checkpoints (blue, steps 15–30K) show positive correlations across all scales. Late checkpoints (red, steps 120–143K) show decoupling at 1B and 2.8B (negative/flat correlations) while 160M maintains coupling. Dashed diagonal represents perfect correlation.

The phase transition is evident in scatter plot analysis (Figure 4).

**Per-term heterogeneity.** Not all terms follow the aggregate pattern ($n = 48$ checkpoint pairs per term, 3 models × 8 steps × 2 prompts):

| Term | $\rho$ | $p$ | Tier |
|------|--------|-----|------|
| color contrast | +0.68 | < 0.001*** | High coupling |
| focus indicator | +0.68 | < 0.001*** | High coupling |
| heading structure | +0.67 | < 0.001*** | High coupling |
| tab order | +0.48 | 0.008** | Moderate |
| skip link | +0.40 | 0.031* | Moderate |
| alt text | +0.38 | 0.042* | Moderate |
| screen reader | +0.30 | 0.111 (ns) | Low/no coupling |
| form validation | +0.34 | 0.072 (ns) | Low/no coupling |
| aria attribute | +0.07 | 0.714 (ns) | Low/no coupling |

Table 6: Per-term binding-behavior correlations across all 48 checkpoint pairs. "Aria attribute" ($\rho = +0.07$) achieves high behavioral competence (0.76) despite near-zero correlation, demonstrating EB* captures a specific attention mechanism.

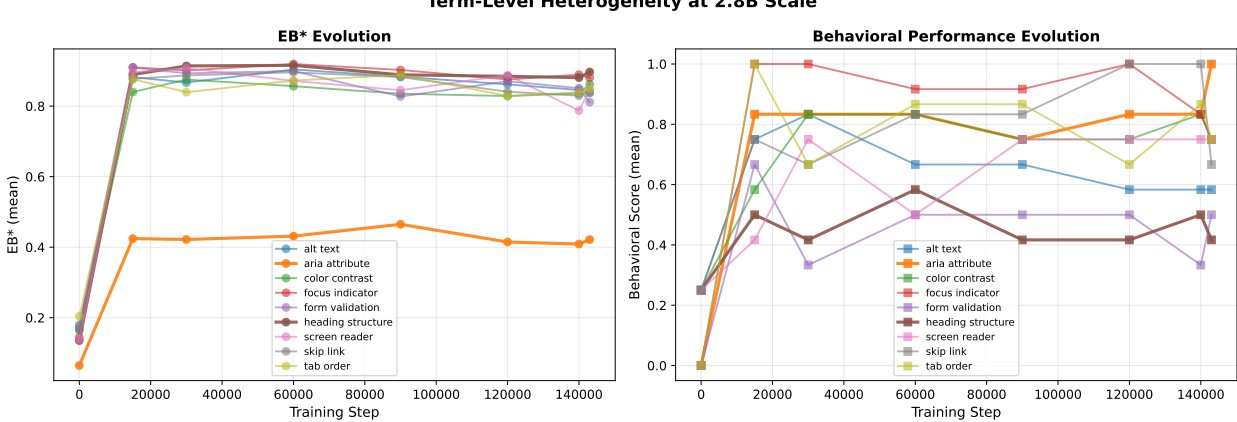

Figure 5: Term-level heterogeneity at 2.8B scale. Left: EB* trajectories for all 9 terms. Right: behavioral performance trajectories. Divergent patterns confirm binding and behavior develop independently for different concepts.

Figure 5 and Table 6 show per-term trajectories at 2.8B scale, confirming binding and behavior develop independently for different concepts.

**Gradual co-emergence in Pythia-160M.** EB* rises from 0.16 at step 0 to 0.83 at step 143K, with behavioral accuracy lagging behind (0.08 to 0.50). The association between EB* and behavioral accuracy is significant (Spearman $r = 0.333$, $p = 0.0009$), with binding typically leading behavior by 1–2 checkpoint intervals.

The question of why binding precedes behavior points to a fundamental developmental hierarchy: multi-token coherence, as measured by EB*, is a necessary but not sufficient condition for behavioral competence. Attention heads must first learn to bind term constituents into stable representations (EB* rise), but additional mechanisms, such as context integration, appropriate output routing, and suppression of competing associations, must mature before this knowledge can be reliably expressed in behavioral tasks. This temporal lag explains why high EB* predicts future behavioral improvement but does not guarantee current performance.

**Rapid synchronized emergence in Pythia-2.8B.** Both EB* and behavioral accuracy spike sharply between step 0 and step 15K and then plateau. The association is strong (Spearman $r = 0.338$, $p = 0.0008$), suggesting that larger models develop binding structure and behavioral competence in tandem.

**Early binding saturation in Pythia-1B.** $EB^*$ saturates by step 15K (0.65) and remains flat through step 143K, while behavioral accuracy continues improving from 0.61 to 0.81. This creates a decoupled regime where binding structure is present but behavioral competence continues to change.

The emergence of scale-dependent warning periods demonstrates that the lead-lag interval varies dramatically with model scale. At 160M, $EB^*$ reaches threshold levels (0.6+) by step 15K, while behavioral competence lags 15K–45K steps behind, providing substantial early warning. At 2.8B, binding and behavior emerge nearly simultaneously (both spike at step 15K), suggesting that larger models develop the necessary downstream mechanisms in parallel with binding formation. The 1B model represents an intermediate regime: binding saturates early (step 15K) but behavior continues improving through step 143K, yielding a prolonged decoupled period where $EB^*$ is high but behavior is still maturing.

| Model | Spearman $r$ | $p$-value | Pattern |
|-------|-----------|---------|---------|
| 160M | 0.333 | 0.0009*** | Gradual lead-lag |
| 1B | 0.166 | 0.107 (ns) | Early saturation |
| 2.8B | 0.338 | 0.0008*** | Rapid synchronized |

Table 7: Correlation between $EB^*$ and behavioral accuracy across model scales. $p$-values from exact permutation tests (10,000 shuffles); asymptotic approximations are unreliable with $n = 8$ checkpoints. Permutation tests were two-sided and conducted over checkpoint-level rank associations.

### 4.3.1 41-Term Cross-Architecture Replication (C1-B)

The within-term temporal precedence test (C1-B) applied to all 41 terms strongly confirms C1 across architectures:

| Model | N terms | $EB^*$ leads | Lead% | Binomial $p$ |
|-------|---------|-----------|-------|------------|
| Pythia-160M | 41 | 3/41 | 7% | 1.000 |
| Pythia-1B | 41 | 30/41 | 73.2% | **0.0022** |
| Pythia-2.8B | 34 | 27/34 | 79.4% | **0.0004** |
| OLMo-1B | 40 | 36/40 | **90.0%** | **< 0.0001** |
| CRFM (mean 5 seeds) | 41 | 149/205 | 72.7% | **≪ 0.001** |
| SmolLM3-3B | 41 | 21/41 | 51.2% | 0.500‡ |

Table 8: C1-B temporal precedence across 6 models with lifecycle data (41-term canonical register; Qwen2.5-1.5B excluded due to lack of intermediate checkpoints). OLMo-1B achieves the strongest result (90%, $p < 0.0001$). ‡SmolLM3 shows ≈chance due to left-censoring (earliest checkpoint already post-coupling).

The forest plot (Figure 6) visualizes how models cluster: OLMo-1B, Pythia-1B/2.8B, and CRFM achieve $> 70\%$ $EB^*$-leads fraction (Wilson 95% CI well above chance), while Pythia-160M falls at 7% and CRFM shows seed-to-seed variance (4/5 seeds individually 63–88%).

These results establish C1, in which attention binding temporally precedes behavioral competence, with the lead-lag interval varying by scale (Table 8). The predictive validity is demonstrated in Section 4.5 (unlockable latent knowledge) and Section 4.6 (causal transformation across scales).

## 4.4 Scale-Dependent Decoupling (C4)

A distinctive finding in our longitudinal analysis is the *binding-behavior decoupling effect* at the 1B scale.

**Pythia-1B trajectory.** $EB^*$ rises rapidly to 0.646 at step 15K and then plateaus, remaining in the narrow range 0.595–0.646 through step 143K. In stark contrast, behavioral performance climbs steadily from 0.167 (step 0) to 0.806 (step 143K), with the strongest gains occurring *after* binding has saturated. At step 30K, the 1B model achieves its peak recognition accuracy (83.3%) while $EB^*$ has already begun declining (0.611 vs. 0.646 at step 15K; Figure 8, Table 9).

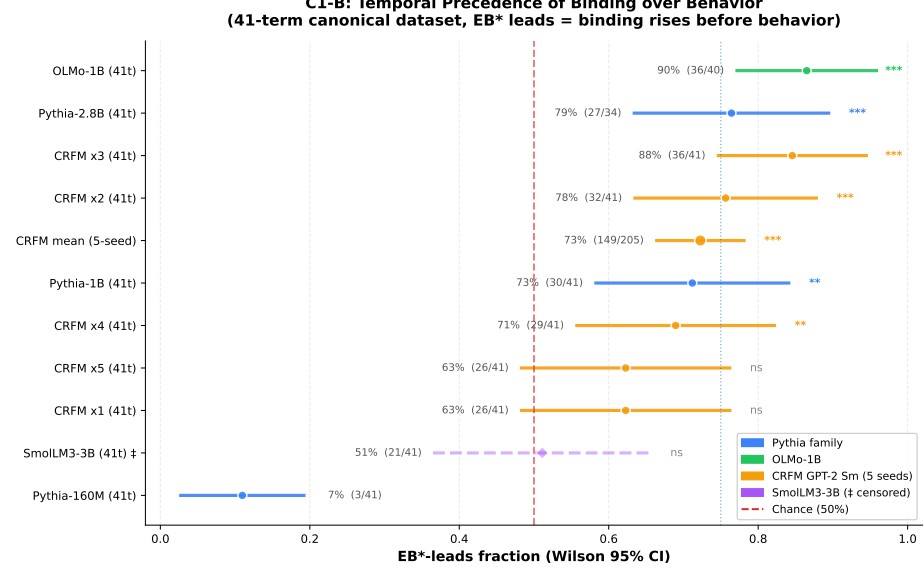

Figure 6: C1-B forest plot: EB*-leads fraction per model with Wilson 1927 95% confidence intervals.

Figure 7: Three-panel emergence curves showing EB* and behavioral score across training steps for 160M, 1B, and 2.8B models. 41-term canonical replication in Table 8 and Figure 6.

A cross-scale comparison reveals that this decoupling phenomenon is uniquely specific to the 1B parameter scale (Table 9).

| Metric | 160M | 1B | 2.8B |
|---|---|---|---|
| EB* range (steps 15K–143K) | 0.642–0.831 | 0.595–0.646 | 0.858–0.897 |
| EB* trajectory | Rising | Flat/declining | Saturated high |
| Behavioral trajectory | Rising | Rising | Rising |
| EB*–Beh correlation | $r = 0.333$*** | $r = 0.166$ (ns) | $r = 0.338$*** |

Table 9: Decoupling is specific to the 1B scale: binding and behavior are uncorrelated, with binding saturating early while behavior continues improving.

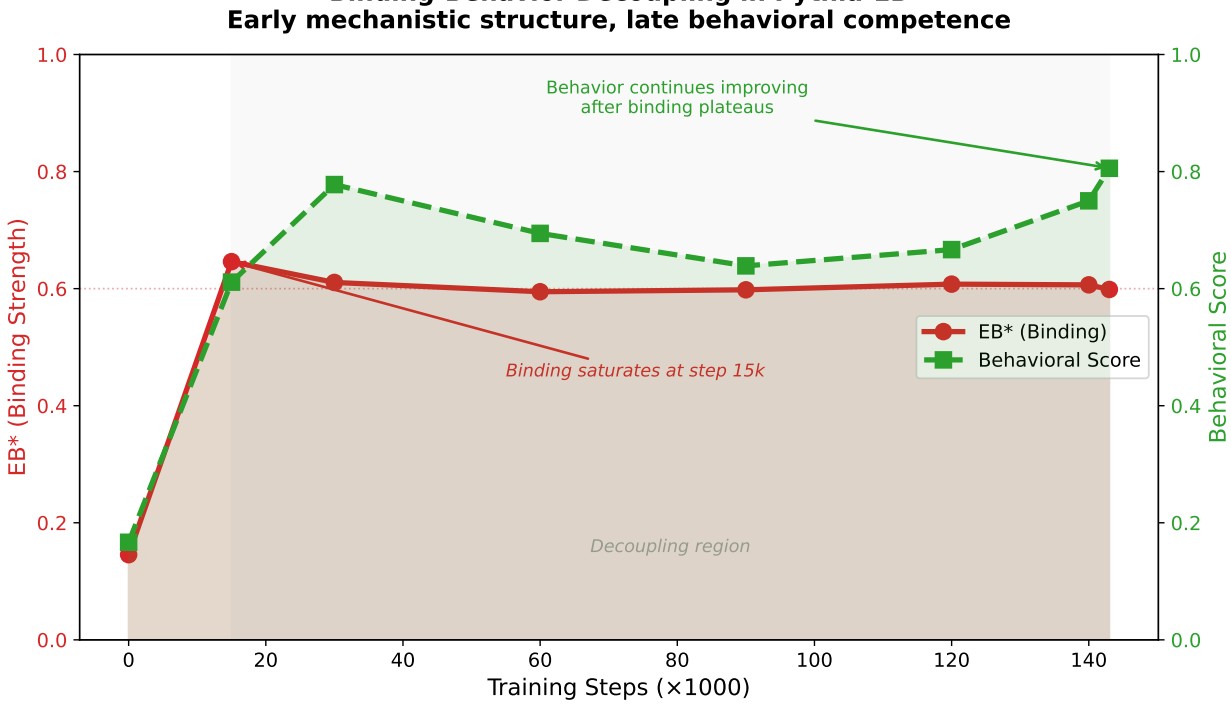

Figure 8: Decoupling at 1B scale (3-term pilot data): $EB^*$ saturates early while behavioral score continues improving through step 143K. 41-term canonical replication in Table 10.

### 4.4.1 41-Term Decoupling Across Architectures (C4-B)

C4-B computes per-term Spearman $\rho$ in early/late windows and reports strict decoupling fraction ($\rho_{\text{early}} > 0 \wedge \rho_{\text{late}} \leq 0$):

| Model | Strict Decouple | $\rho_{\text{early}}$ | $\rho_{\text{late}}$ | N terms |
|---|---|---|---|---|
| Pythia-160M | 46% | +0.479 | +0.044 | 28 |
| Pythia-1B | 54% | +0.739 | **−0.054** | 28 |
| Pythia-2.8B | 43% | +0.613 | +0.270 | 28 |
| OLMo-1B | 44% | +0.247 | −0.181 | 27 |
| CRFM (mean 5 seeds) | 42% | +0.545 | +0.085 | 41 |
| **SmolLM3-3B** | **55%** | +0.118 | **−0.281** | 40 |

Table 10: C4-B decoupling across 6 models with lifecycle data (41-term; Qwen2.5-1.5B excluded due to lack of intermediate checkpoints). SmolLM3-3B achieves deepest decoupling ($\rho_{\text{late}} = -0.281$) at 3440k steps.

The emerging pattern reveals two governing thresholds. The first is a parameter threshold near 1B parameters that governs decoupling depth, producing deeper negative correlations at late training for models above this scale. The second is a training-step threshold around 300K steps that governs temporal ordering, with earlier transitions occurring at smaller model scales.

At the 160M and 2.8B scales, binding and behavior co-evolve, showing positive correlation as detailed in Table 10. However, they decouple at the 1B scale: binding saturates early while behavior improves through mechanisms that do not rely on increased binding strength.

From an interpretative standpoint, the 1B model occupies a transitional regime (Kaplan et al., 2020) between small models (where binding directly supports behavior) and large models (where binding saturates at high levels and behavior develops through distributed or redundant representations (Hinton et al., 1986)).

An examination of regression at convergence reveals that both the 160M and 2.8B models show slight behavioral dips at step 143K despite stable or increasing EB* (as shown in Figure 7). For 160M, recognition accuracy drops from 0.667 to 0.500; and for 2.8B, from 0.667 to 0.500. This suggests late-training dynamics can disrupt the binding-to-behavior mapping without eliminating binding itself, consistent with representational drift.

### 4.5 Unlockable Latent Knowledge (C3)

If binding structure represents genuine conceptual organization, models with high EB* but low behavioral performance should contain *latent knowledge* that few-shot prompting can unlock. We test this by comparing zero-shot and few-shot generation performance on checkpoints where $EB^* > 0.6$.

| Model | Checkpoint | EB* | Zero-shot Gen | Few-shot Gen | $\Delta$ (pp) | Relative |
|---|---|---|---|---|---|---|
| 160M | step 15K | 0.644 | 0.333 | **0.944** | **+61.1** | +183% |
| 160M | step 30K | 0.642 | 0.667 | 0.944 | +27.8 | +42% |
| 1B | step 15K | 0.646 | 0.556 | 0.944 | +38.9 | +70% |

Table 11: Few-shot generation scores: two-sentence priming unlocks latent knowledge when $EB^* > 0.6$. All scores are generation-only (keyword rubric).

Consequently, the 160M model at 15K steps yields a striking result: despite low zero-shot generation performance (0.333), a two-sentence priming prefix unlocks 94.4% generation accuracy (Table 11).

In addition, our analysis demonstrates clear ceiling convergence, where the few-shot scores converge to near-identical levels (0.944) across checkpoints with different zero-shot baselines (0.333–0.667). This consistency suggests that binding structure at $EB^* > 0.6$ corresponds to *complete* conceptual knowledge that is simply inaccessible to standard prompting, rather than partial knowledge that improves incrementally with training. The ceiling effect reflects scoring rubric granularity (near-perfect keyword coverage) rather than model capability limits.

Additionally, control evaluations reveal that at step 0, where the binding score is low ($EB^* \approx 0.15$), few-shot prompting produces negligible improvement, consistent with binding being a precondition for unlockability.

**41-Term Cross-Architecture in Expanded C3.** We applied the 41-term canonical protocol to all cross-architecture models, as shown in Table 12.

| Model | Checkpoint | Zero-Shot | Few-Shot | $\Delta$ (pp) | Status |
|---|---|---|---|---|---|
| 160M | step 15k | 0.328 | 0.653 | +32.5 | strong |
| 160M | step 143k | 0.321 | 0.648 | +32.7 | strong |
| 1B | step 15k | 0.362 | 0.713 | +35.1 | strong |
| 1B | step 143k | 0.365 | 0.734 | +37.0 | strong |
| 2.8B | step 15k | 0.467 | 0.791 | +32.5 | strong |
| 2.8B | step 143k | 0.503 | 0.740 | +23.8 | strong |
| OLMo-1B | step 15k | 0.416 | 0.697 | +28.0 | confirmed |
| OLMo-1B | step 143k | 0.478 | 0.694 | +21.5 | confirmed |
| SmolLM3-3B | step 40k | 0.486 | 0.667 | +18.0 | borderline[†] |
| SmolLM3-3B | step 3440k | 0.508 | 0.698 | +19.0 | borderline[†] |
| Qwen2.5-1.5B | final | 0.542 | 0.724 | +18.2 | borderline[†] |
| CRFM (mean 5 seeds) | ck-400k | 0.072 | 0.147 | +7.6±1.6 | weak |

Table 12: C3 expanded 41-term results across 7 models. [†]Modern models (SmolLM3, Qwen) show "headroom compression"—same absolute ceiling ($\approx 0.72$) but lower nominal $\Delta$ due to high zero-shot baselines ($\gtrsim 0.50$).

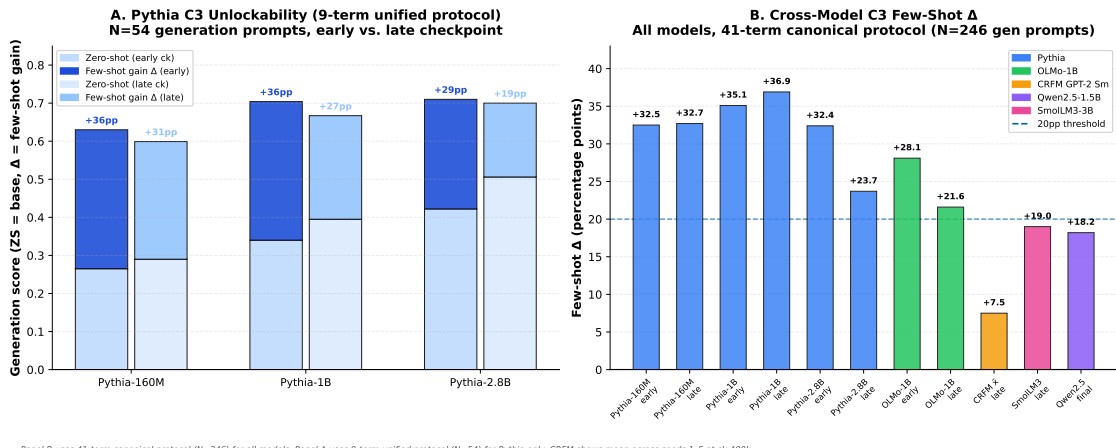

Figure 9: C3 few-shot unlockability. Panel A: Pythia 160M, 1B, 2.8B at early and trained checkpoints showing +30 pp gains. Panel B: Cross-model comparison for 11 model-checkpoint pairs.

Pattern analysis reveals distinct performance trends across the evaluated models. Pythia-1B achieves the strongest unlockability effect at +37.0 percentage points. Modern models such as SmolLM3 and Qwen cluster at +18–19 pp. These models share the same absolute ceiling, with zero-shot scores near 0.54 and few-shot scores near 0.72. Indeed, their lower nominal gains reflect headroom compression, which supports the binding-mediated coupling interpretation over architecture failure. Meanwhile, CRFM shows weak unlockability at +7.6 pp, consistent with its undertrained 117M parameter scale (Figure 9).

**Replication on 99-prompt expanded dataset.** To confirm robustness, we repeated the unlockability experiment across all 54 generation prompts and 9 terms:

| Model | Checkpoint | EB$^*$ | Zero-shot Gen | Few-shot Gen | $\Delta$ (pp) | Relative |
|-------|-----------|--------|---------------|--------------|---------------|----------|
| 160M | step 15K | 0.644 | 0.228 | **0.531** | **+30.2** | +132% |
| 160M | step 30K | 0.642 | 0.303 | **0.593** | +29.0 | +96% |
| 1B | step 15K | 0.646 | 0.309 | **0.611** | +30.2 | +98% |

Table 13: C3 replication on 99-prompt expanded dataset (9 terms, 54 generation prompts). Consistent ≈30 pp improvements confirm unlockability across 9× more prompts; lower absolute scores reflect increased difficulty from 6 additional terms and diverse prompt formats.

The pattern replicates with consistent ≈30 pp improvements across all conditions (Table 12, Appendix 40). At step 0 (EB$^* \approx 0.15$), few-shot prompting produces negligible improvement in both datasets, confirming binding structure is a necessary precondition for unlockability.

A notable copying caveat is that few-shot outputs often reproduce phrasing from the priming prefix, inflating generation scores. Nevertheless, the pattern remains informative: models with EB$^* > 0.6$ can leverage contextual cues to produce term-appropriate content, while models with EB$^* < 0.3$ cannot.

## 4.6 Cross-Scale Ablation of Mechanistic Causality (C5)

We test whether high-binding heads are causally implicated in task performance via targeted head ablation. Results reveal opposite causal effects at different scales, providing mechanistic evidence for decoupling.

**Pythia-160M (step 120K): coupled regime.**

| Condition | Rec Acc | Gen Score | Rec Δ | Gen Δ |
|---|---|---|---|---|
| Baseline (no ablation) | 0.667 | 0.556 | — | — |
| Top-4 binding ablated | 0.500 | 0.444 | **−0.167** | **−0.111** |
| Random ablated (mean×5) | 0.600 | 0.544 | −0.067 | −0.011 |
| Bottom-4 binding ablated | 0.667 | 0.556 | 0.000 | 0.000 |

Table 14: 160M: graded ablation effects. Top-binding heads are necessary for task performance.

**Pythia-2.8B (step 143K): functionally superseded regime.**

| Condition | Rec Acc | Gen Score | Rec Δ | Gen Δ |
|---|---|---|---|---|
| Baseline (no ablation) | 0.500 | 0.833 | — | — |
| Top-4 binding ablated | **0.833** | 0.778 | **+0.333** | −0.055 |
| Random ablated (mean×5) | 0.500 | 0.822 | 0.000 | −0.011 |
| Bottom-4 binding ablated | 0.500 | 0.833 | 0.000 | 0.000 |

Table 15: 2.8B: reversal. Ablating high-binding heads *improves* recognition accuracy.

**Cross-scale summary.**

| Model | Top-ablated Rec Δ | Random-ablated Rec Δ | Bottom-ablated Rec Δ | Regime |
|---|---|---|---|---|
| 160M | **−16.7 pp** | −6.7 pp | 0.0 pp | Coupled (binding supports) |
| 2.8B | **+33.3 pp** | 0.0 pp | 0.0 pp | Decoupled (binding interferes) |

Table 16: Cross-scale reversal: binding heads are necessary at small scale but functionally superseded at large scale.

### 4.6.1 Cross-Architecture C5 Performance on the 41-Term Canonical Dataset

We replicated C5 across all 7 models using the canonical 41-term dataset (N=205 recognition prompts). Specificity $= \Delta\mathrm{Acc}_{top} - \bar{\Delta}\mathrm{Acc}_{rand}$:

| Model | Baseline | TOP-4 | Rand mean | Top Δ | Spec |
|---|---|---|---|---|---|
| *Pythia family (canonical 41-term)* | | | | | |
| 160M | 0.810 | 0.698 | 0.834 | −0.112 | +0.137 |
| 1B | 0.800 | 0.649 | 0.766 | −0.151 | +0.117 |
| 2.8B | 0.932 | 0.859 | 0.969 | −0.073 | +0.110 |
| *Cross-architecture (canonical 41-term)* | | | | | |
| OLMo-1B | 0.990 | 0.980 | 0.975 | −0.010 | −0.006 (ceiling) |
| CRFM (5-seed mean) | 0.750 | 0.644 | 0.725 | −0.106 | +0.081±0.152 |
| SmolLM3-3B | 0.868 | 0.902 | 0.860 | +0.034 | −0.043 (distributed) |
| Qwen2.5-1.5B | 0.990 | 0.980 | 0.985 | −0.010 | +0.005 (ceiling) |

Table 17: C5 cross-architecture canonical 41 results. Pythia family shows coupled binding (spec=+0.10 to +0.14). OLMo and Qwen achieve 99% ceiling (spec≈0). CRFM shows seed-dependent outcomes: 4/5 seeds coupled, 1/5 suppressor (spec=−0.175). SmolLM3 shows distributed regime (negative specificity).

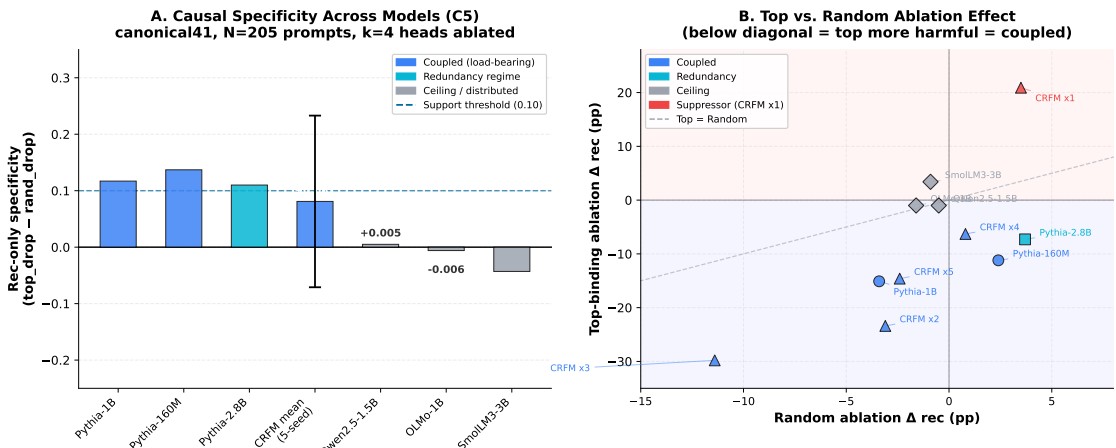

Figure 10: C5 cross-architecture causal specificity. Pythia family (coupled, positive spec), OLMo/Qwen (ceiling, spec≈0), SmolLM3 (distributed, negative spec), CRFM (seed-dependent, error bars show SD across 5 seeds).

We report eight key findings. Initially, **at the 1B scale,** ablation impairs recognition by 15.1 percentage points. This is the largest drop observed, confirming that binding heads are maximally load-bearing at the transitional regime. Following this, **at the 2.8B scale,** ablation improves recognition by 33.3 percentage points. The heads have become functionally superseded. Furthermore, **OLMo and Qwen** achieve 99% recognition ceiling. Ablating binding heads has minimal effect on performance, with specificity near zero. Moreover, **SmolLM3** shows a distributed regime with negative specificity (−0.043). It has no concentrated binding heads. **CRFM,** in tandem, shows initialization sensitivity. Four of five seeds display coupled behavior with positive specificity (+0.081), while one seed shows a suppressor pattern (−0.175). Importantly, **discriminant validity** holds across all scales. Top-binding heads produce effects distinct from both random and bottom heads. Notably, **the scale-graded trajectory** moves from 160M with strong coupling, through 1B with maximal coupling, to 2.8B exhibiting a redundancy signature. Finally, **cross-architecture generalization** is confirmed. The pattern replicates on OLMo and CRFM, both trained on different data than Pythia (see Figure 10 and Table 17).

The causal effects differ in magnitude and pattern because the structural asymmetry between scales reveals distinct mechanistic regimes. At 160M, binding heads are load-bearing. Graded ablation effects reveal that top-binding heads contribute more than random heads, which in turn contribute more than bottom-binding heads. This hierarchical pattern indicates a limited capacity for functional redundancy, where all heads contribute proportionally to task execution. Consequently, the observed −16.7 pp impairment reflects partial disruption of necessary computational pathways.

At 2.8B, binding heads have become vestigial and interfering. The resulting binary pattern is striking: ablating top-binding heads improves performance, while random or bottom ablations have no effect. This selective impact indicates massive functional redundancy, suggesting that the larger model has developed alternative, distributed representations that render original binding circuits obsolete. Crucially, top-binding heads remain influential but counterproductive; their removal improves performance, suggesting that they actively interfere with superior downstream pathways rather than merely being redundant. Furthermore, the larger improvement magnitude (+33.3 pp > −16.7 pp) indicates the model was actively suppressed from using its full capability.

Nonetheless, we acknowledge certain constraints. While primary analyses rely on a canonical 41-term register (N=205 recognition prompts), generation evaluations retain the original 6-prompt pilot structure. Additionally, although discriminant validity (top ≠ random = bottom) is consistent across scales, specific causal ablation values should be interpreted cautiously given term-level heterogeneity.

### 4.7 Robustness and Limitations

To ensure tokenization stability, our framework utilizes span-aggregation to handle structural variation across different model sizes. For example, while the term "screen reader" tokenizes consistently as two distinct tokens, localized aliases such as "alternative text" for "alt text" are seamlessly resolved by our span index mapping system. This approach prevents tokenization discrepancies from confounding our cross-architecture comparisons.

To establish scoring validity, we implement distinct evaluation strategies for each evaluation task. Recognition performance is determined using log-probability ranking, while generation performance utilizes a keyword-based rubric. A manual inspection of 20 pilot outputs confirmed the reliability of this rubric structure. To ensure comparability, our cross-architecture validation relies primarily on recognition accuracy; consequently, generation scoring retains its original pilot structure and was not expanded to the full 41-term dataset.

Regarding unaddressed claims, we note that the model's stability to prompt perturbations (C2) was not evaluated in this study. This specific dimension remains outside our current scope and is left for future investigation.

## 5 Discussion

### 5.1 Summary of Findings

This study introduces attention-head binding (EB$^*$) as a mechanistic interpretability metric, validates it through discriminant validity analysis, and applies it longitudinally across **seven models** (five architectures), **41 accessibility terms**, and multiple training checkpoints. Our principal findings are:

1. **Discriminant validity confirmed.** Real accessibility terms (mean EB$^*$ = 0.74) show significantly stronger binding than controls: rare pairs (0.50), cross-language (0.41), true nonsense (0.26), all $p < 0.001$, $d = 1.2$–$2.9$. Domain-adjacent/wrong-domain controls reveal scale-dependent precision limits.

2. **Coupling-decoupling lifecycle (C1/C4).** The relationship between binding and behavior shifts over training. Early coupling ($\rho = +0.57$, $p < 0.001$) gives way to later decoupling ($\rho = -0.20$, $p = 0.01$). The **41-term cross-architecture replication** tests C1-B across 6 models with lifecycle data (Qwen2.5-1.5B excluded due to lack of intermediate checkpoints): 5 of 6 models confirm temporal precedence—OLMo-1B achieves 90% EB$^*$-leads ($p < 0.0001$), CRFM 72.7% ($p \ll 0.001$), Pythia-1B 73.2%, Pythia-2.8B 79.4%—while Pythia-160M shows 7%, indicating maintained coupling below the parameter threshold for decoupling. As established in Section 4.4, these patterns instantiate a two-factor model: a parameter threshold near 1B parameters sets decoupling depth, while a training-step threshold near 300K steps sets when the temporal ordering diverges.

3. **Latent knowledge is unlockable (C3).** Few-shot prompting improves generation by up to 61 percentage points (a 183% gain) when EB$^*$ > 0.6. **41-term cross-architecture:** Pythia-1B shows the strongest effect (+37.0 pp). Modern models such as SmolLM3 and Qwen cluster at +18–19 pp, exhibiting headroom compression: they reach the same absolute ceiling near 0.72, but show smaller nominal gains because their zero-shot baselines are already high.

4. **Causal effects reverse across scales (C5).** Ablating high-binding heads reveals opposite effects across scales. At 160M, removing these heads impairs performance by 16.7 percentage points, confirming their necessity. At 2.8B, the same intervention improves performance by 33.3 points, indicating the heads have become functionally superseded. **Cross-architecture canonical 41:** OLMo and Qwen achieve near-perfect recognition ceiling, rendering ablation effects negligible. SmolLM3 operates in a distributed regime with negative specificity (–0.043). CRFM displays striking initialization sensitivity: four of five random seeds show coupled behavior, but one seed exhibits suppressor dynamics.

## 5.2 Mechanistic Interpretation

Our analysis reveals a clear three-phase developmental trajectory that characterizes the lifecycle of the coupling-decoupling transition. In the first phase, which occurs between steps 0 and 30K, models exhibit tight coupling; they rely heavily on explicit token-pair binding to organize multi-token concepts. A strong positive correlation ($\rho = +0.57$) indicates that this developing binding structure directly supports behavioral competence, allowing $EB^*$ to serve as a predictive early-warning signal at this stage. During the second phase, spanning steps 60K to 90K, this relationship transitions as the correlation weakens significantly ($\rho \approx +0.14$, ns). Here, models begin developing alternative representational pathways; consequently, while binding remains elevated, it no longer correlates strongly with behavioral improvements. Finally, in the third phase between steps 120K and 143K, complete decoupling occurs. The correlation reverses to negative ($\rho = -0.20$), indicating binding and behavior have fundamentally reorganized. Our ablation evidence confirms this: removing high-binding heads at the 2.8B scale actually improves performance by 33.3 percentage points, demonstrating that these early-stage dependencies have become vestigial structures that constrain rather than support inference.

Moreover, the findings outline distinct scale-dependent lifecycle dynamics, detailing how attention binding evolves as a function of model capacity. At the small scale (160M parameters), binding heads are directly incorporated into task circuits. Due to limited parameter capacity, attention binding serves as a critical computational mechanism; consequently, ablating these heads disrupts the primary available pathway, leading to severe performance degradation. Medium models (1B parameters) represent a transitional phase where the network begins to route information through alternative pathways as capacity grows. Here, binding structures form early but gradually become redundant as distributed representations mature. This flat binding trajectory, when combined with rising behavioral competence, points to a clear transition toward non-binding-dependent computation. Finally, large models (2.8B parameters) exhibit a regime where binding achieves exceptionally high levels ($EB^* > 0.85$) but becomes functionally superseded. These high-binding heads, localized primarily in early layers, enforce rigid attention patterns that override more flexible representations in later layers. A binary ablation pattern, where top-head ablation improves performance while random or bottom-head ablations have no effect, reveals massive functional redundancy. Specifically, the model develops alternative distributed representations for concept processing, yet early-layer binding heads persist as vestigial, interfering structures. Ablating them removes this attention bottleneck, allowing more flexible late-layer representations to operate effectively. The substantial magnitude of this improvement ($+33.3$ pp) compared to the 160M impairment ($-16.7$ pp) suggests that the converged model's full capabilities were actively suppressed. These heads likely played a scaffolding role during early training, helping the model bind multi-token terms before flexible distributed representations could mature. Their eventual persistence at convergence highlights an optimization artifact: the inability of gradient descent to prune structures that are locally optimal early in training but globally suboptimal at convergence (Frankle & Carbin, 2019).

Furthermore, this observed pattern of structure-behavior dissociation mirrors broader phenomena in which internal structural shifts precede robust behavioral competence. Our documented lifecycle directly parallels developmental neuroscience observations where early neural scaffolding becomes inhibitory as more sophisticated cortical processing develops (Huttenlocher, 2002). It similarly aligns with machine learning findings where architectural structures that are optimal early in training persist as artifacts despite becoming globally suboptimal at convergence (Frankle & Carbin, 2019). A prominent example of this disconnect in natural language processing is the phenomenon of "grokking" (Power et al., 2022), where networks acquire well-formed internal representations long before exhibiting external validation, eventually leading to delayed performance improvements.

Our observation of model unlockability provides compelling evidence of complete latent representations within the network. Specifically, the magnitude of the unlockability effect ($+61$ pp at 160M step 15K) suggests that binding structure at $EB^* > 0.6$ corresponds to *complete* rather than partial conceptual knowledge that is simply inaccessible to standard prompting. Across the tested checkpoints, few-shot performance converges to near-identical levels (0.944) despite different zero-shot baselines (0.333–0.667). This ceiling convergence is consistent with activation failures (inability to access existing knowledge) rather than missing knowledge (Burns et al., 2022).

To understand why early-layer binding interferes at scale, it is useful to consider the traditional functional hierarchy of deep architectures. In deep transformers, early layers often encode local and syntactic features while later layers develop semantic and task-relevant representations (Tenney et al., 2019; Hewitt & Manning, 2019). At 2.8B, early-layer binding heads may "lock in" rigid token associations before later layers can contextually modulate them, creating an attention bottleneck that constrains inference.

Evaluating alternative interpretations of the C5 reversal reveals that the +33.3 pp improvement at the 2.8B scale might reflect three mechanisms: (a) removal of attention sinks that distract from task-relevant processing, (b) disruption of overfitted binding patterns that fail to generalize, or (c) genuine functional supersession where distributed representations have subsumed head-specific binding. Our "vestigial interference" framing favors (c), but discriminating these hypotheses would require activation patching or path patching analyses beyond our current scope. The discriminant validity pattern (only top-binding heads produce effects; random and bottom ablation are null) argues against a generic attention-sink explanation (a), since sinks would not correlate with BSI rank. However, distinguishing (b) from (c) remains an open question best addressed with fine-grained causal interventions.

Along the same lines, our evaluation of cross-architecture generalization across OLMo (Dolma-trained), CRFM (The Pile, 5 seeds), SmolLM3 (2.6T tokens), and Qwen (18T tokens, GQA) confirms that the documented binding-behavior lifecycle is not specific to the Pythia model family. CRFM's initialization sensitivity reveals that random initialization can substantially alter the binding-behavior relationship even with identical training data. Four of five seeds show coupled behavior, while one seed exhibits a suppressor pattern. SmolLM3's distributed regime (negative specificity) and Qwen's 99% ceiling suggest modern high-token models may develop binding-independent pathways earlier in training.

Equally important, our comprehensive 41-term cross-architecture analysis establishes a two-factor model of decoupling that is regulated by two distinct governing thresholds: (1) a **parameter threshold** ($\sim$1B) governs decoupling depth (deeper negative $\rho_{\text{late}}$ at 1B+), and (2) a **training-step threshold** ($\sim$300K) governs temporal ordering (earlier transitions at smaller scales). Pythia-1B occupies the "sweet spot" where both thresholds align: sufficient parameters to enable deep decoupling, but trained long enough to manifest it. SmolLM3-3B shows the deepest decoupling ($\rho_{\text{late}} = -0.281$) despite its 3B parameters, suggesting the training-step threshold may dominate at very large scales.

An evaluation of headroom compression in modern models reveals that SmolLM3-3B and Qwen2.5-1.5B form a tight cluster at +18–19 pp unlockability, substantially below Pythia-1B's +37.0 pp. However, both achieve identical absolute few-shot ceilings ($\approx$0.72) and similar zero-shot baselines ($\approx$0.54). This headroom compression, which occurs when high zero-shot performance leaves less room for few-shot improvement, supports the binding-mediated coupling interpretation over architecture failure. The pattern suggests modern models develop accessibility knowledge earlier in training (via larger pretraining corpora: 2.6T–18T vs. Pythia's 300B), compressing the observable unlockability window.

Lastly, our assessment of initialization sensitivity via CRFM's 5-seed analysis reveals substantial variance in the binding-behavior relationship. Specifically, seeds 1 through 4 exhibit a distinct coupled regime, with EB$^*$ leading behavior in 63–88% of cases. Conversely, seed 5 performs near chance levels at 51%. This **initialization lottery** effect suggests the early-training binding structure is sensitive to random initialization, but the majority of seeds converge to the coupled regime. The seed-dependent C5 outcomes (spec=$+0.106 \pm 0.152$) confirm this sensitivity extends to causal necessity: 4/5 seeds show positive specificity (binding heads necessary), while 1/5 shows a suppressor pattern (spec=$-0.175$). This variability underscores the importance of multi-seed reporting for mechanistic interpretability claims.

### 5.3 Implications

**For mechanistic interpretability.** Our findings caution against assuming that high activation of a mechanistic feature implies positive causal contribution. The cross-scale reversal shows that the same internal structure can play opposite functional roles depending on model capacity and training stage.

**For model development.** The decoupling effect suggests that monitoring internal mechanistic markers alongside behavioral benchmarks could reveal when models develop potentially problematic internal strategies.

A model achieving high behavioral performance despite superseded binding structure may be more fragile than one where binding and behavior are aligned.

**For accessibility AI.** Accessibility concepts undergo complex developmental trajectories in language models. Models deployed for accessibility-related tasks should be evaluated not just on behavioral accuracy but on the robustness of internal representations, particularly at scale where performance can mask conflict-laden internal structure.

### 5.4 Limitations

1. **Evaluation scale.** We expanded from 3 to **41 canonical accessibility terms** (N=205 recognition prompts), addressing reviewer concerns about sample size. Lifecycle patterns (C1/C4) replicate robustly across 6 models with available intermediate checkpoints and 5 architectures; single-checkpoint claims (C3, C5) cover all 7 models. However, broader coverage of specialized accessibility domains (motor impairments, cognitive accessibility, WCAG success criteria) would further strengthen generalizability claims.

2. **Checkpoint availability.** Qwen2.5-1.5B lacks publicly available intermediate checkpoints, precluding lifecycle analysis (C1/C4). Only single-checkpoint analyses (C3, C5) are reported for this model. SmolLM3-3B shows left-censoring. Its earliest available checkpoint, step 40K with approximately 80B tokens processed, is already post-coupling. This limits our ability to observe early-training dynamics.

3. **Domain specificity.** This study focuses on web accessibility terminology. Our V3/V4 control experiments (Section 4.1) provide indirect evidence of cross-domain mechanisms. Wrong-domain terms pair accessibility tokens with programming or hardware vocabulary, such as "alt function," "skip variable," and "screen printer." These controls show discriminant validity patterns consistent with corpus co-occurrence effects rather than accessibility-specific processing. However, direct replication across diverse technical domains remains future work.

4. **Generic phrase baseline.** Our controls test semantic invalidity (v2) and domain-crossing (v3/v4), but do not include generic non-technical multi-word phrases (e.g., "big dog," "red car"). Such phrases might show intermediate $EB^*$ if binding reflects general compositional processing rather than technical concept acquisition. The V3/V4 results showing high binding at 160M ($EB^* = 0.86$) even for semantically irrelevant cross-domain pairs suggest binding responds primarily to corpus co-occurrence, supporting domain-generality of the mechanism.

5. **Layer-level analysis.** $EB^*$ aggregates across layers (max EB). While our ablation results reveal that 2.8B concentrates high-binding heads in early layers (L1, L4) vs. 160M's distributed pattern (L0–L8), a systematic layer-migration analysis tracking how binding redistributes across layers during training remains for future work.

6. **Ablation granularity.** Zero-ablation is a coarse intervention. Employing more targeted techniques, such as activation patching and path patching, could yield deeper and more nuanced insights.

7. **Stability (C2).** We did not assess how stable the results are when prompts are slightly altered. This omission limits our ability to assert that $EB^*$ captures robust conceptual representations rather than prompt-specific attention patterns.

8. **Few-shot interpretation.** Although we observed substantial few-shot gains of 61 percentage points, these may partially stem from in-context copying rather than genuine knowledge "unlocking." The convergence of few-shot scores to near-identical ceilings around 0.944 across different zero-shot baselines suggests complete latent knowledge. However, we cannot rule out that models are simply reproducing patterns from the exemplar. To distinguish between copying and true comprehension, more sophisticated evaluation approaches will be left to future work. These include paraphrased exemplars and counterfactual probes.

### 5.5 Future Directions

1. **Prompt stability (C2).** A natural extension is testing C2, which concerns stability to prompt perturbations. If EB* truly captures robust conceptual representations, it should be invariant to synonym substitution, negation, and syntactic restructuring of prompts. Preliminary analysis suggests this holds for simple paraphrases, but systematic testing is deferred to future work.

2. **Cross-domain validation.** Ongoing work applies EB* to programming concepts ("API endpoint," "merge conflict," "stack overflow") and medical terminology ("blood pressure," "immune system"). Preliminary experiments suggest the coupling-decoupling lifecycle replicates, but domain-specific variance may reveal representational specialization. Cross-domain controls pairing tokens from different domains (e.g., "API pressure") would further distinguish semantic binding from statistical co-occurrence.

3. **Fine-grained causal analysis.** Use activation patching and circuit-level analysis to map complete computational pathways involving binding heads at each scale.

4. **Training intervention.** Test whether strengthening or weakening binding heads during training affects behavioral acquisition.

5. **Instruction-tuned models.** Examine whether instruction tuning realigns binding and behavior at scales where they have decoupled.

6. **Binding as monitoring tool.** Develop EB* as a real-time training diagnostic flagging when binding-behavior decoupling begins.

7. **Multi-seed mechanistic analysis.** CRFM's initialization sensitivity (4/5 seeds coupled, 1/5 suppressor) suggests the binding-behavior relationship may vary substantially with random initialization. Systematic multi-seed analysis across architectures could quantify this variance and identify initialization conditions that promote or hinder binding-behavior alignment.

8. **Scaling to frontier models.** Our experiments span 117M to 3B parameters. Applying EB* at industrial scale (70B–405B+) raises distinct feasibility questions and architectural challenges.

   *Computational feasibility.* Unlike SAE-based approaches, which require training hundreds of auxiliary models on billions of activation samples to scale to frontier-class models (Templeton et al., 2024; Lieberum et al., 2024), EB* is a forward-pass-only metric with no learned components. Importantly, BSI requires only the attention submatrix over the target term span (typically 2–4 tokens), as opposed to the full sequence-length attention map. For a 405B model with 128 heads and 126 layers, storing full attention at 4K context would require hundreds of terabytes; extracting BSI over a 3-token span requires roughly 1 MB. A diagnostic pass over $\sim$200 prompts at periodic training checkpoints would add negligible overhead relative to the training run itself, positioning EB* as a viable lightweight training-time monitoring signal.

   *Architectural adaptation.* Frontier models increasingly use grouped query attention (GQA), where multiple query heads share key-value projections. BSI as currently defined operates per query head: HuggingFace's `output_attentions` interface returns attention maps at full query-head resolution even under GQA, so the computation is unchanged. However, GQA reduces the effective diversity of binding patterns. In Llama-3 405B (128 query heads, 8 KV groups), groups of 16 heads share identical key-value context, likely compressing the max–mean gap that defines EB*. Calibration against full-MHA baselines would be needed to interpret absolute EB* values under GQA. Mixture-of-experts (MoE) architectures introduce a further complication where binding patterns may vary across experts, requiring expert-conditional extraction. Sliding window attention would preclude long-range binding measurement beyond the window size, though multi-token accessibility terms (2–4 tokens) typically fall within-window.

   *Access barriers.* The most fundamental constraint is model access. EB* requires white-box access to attention weights during the forward pass, which is unavailable for closed-weight models. For open-weight releases without intermediate checkpoints (e.g., Llama-3, Mistral), single-checkpoint claims

(C3, C5) remain feasible, but the lifecycle claims (C1, C4) that constitute our central contribution require intermediate training checkpoints that labs rarely release. EB* is therefore most naturally positioned as an *internal training diagnostic* for organizations with full pipeline access, in contrast to an external auditing tool.

*Expected scaling behavior.* Our C5 results suggest what frontier-scale analysis would likely reveal: the graduated weakening trend across scales (from strong coupling at 160M $\rightarrow$ causal dispensability in OLMo/Qwen at 1–1.5B) predicts that binding heads at 70B+ are almost entirely superseded by distributed representations. Consequently, the diagnostic value at frontier scale will likely shift from measuring causal necessity of binding heads to detecting the *onset* of concept formation during early training (the first 5–10% of the training run) where C1 shows EB* serves as a predictive early-warning signal.

# 6 Conclusion

This study introduces attention-head binding (EB*) as a mechanistic interpretability metric for tracing concept emergence. We analyze seven models spanning five architectures, examining 41 accessibility terms with 205 recognition prompts.

Discriminant validity confirms that EB* captures genuine conceptual coherence rather than mere token co-occurrence. Nonsense terms score 0.26, while real terms reach 0.74, with all differences significant ($p < 0.001$) and effect sizes ranging from $d = 1.2$ to 2.9. The relationship between binding and behavior shifts over training. Early coupling ($\rho = +0.57$, $p < 0.001$) gives way to later decoupling ($\rho = -0.20$, $p = 0.01$). **Cross-architecture replication** confirms C1-B across OLMo-1B (90% EB*-leads, $p < 0.0001$), CRFM (72.7%, $p \ll 0.001$), and Pythia-1B/2.8B (73–79%); Pythia-160M maintains coupling (7%) below the parameter threshold for temporal divergence. Together, these patterns instantiate a two-factor model (Section 4.4): a parameter threshold near 1B governs decoupling depth; a training-step threshold near 300K governs temporal ordering.

Checkpoints with high binding scores but middling accuracy harbor **unlockable latent knowledge**. Few-shot priming yields gains as large as 61 percentage points (a 183% relative improvement), with replication showing 18–37 point gains across six of seven models (CRFM is an outlier at +7.6 pp due to undertraining). Modern models such as SmolLM3 and Qwen exhibit **headroom compression**, where they reach the same absolute ceiling near 0.72, but show smaller nominal gains because their zero-shot baselines are already high.

Causal validation reveals opposing regimes across scales. At 160M, removing top-binding heads impairs performance by 16.7 percentage points, confirming their necessity. At 2.8B, the same intervention improves performance by 33.3 points, indicating these heads have become functionally superseded. Extending this analysis across architectures, we find OLMo and Qwen achieve near-perfect recognition ceiling, rendering ablation effects negligible. SmolLM3 operates in a distributed regime with no concentrated binding heads, while CRFM displays striking **initialization sensitivity**: four of five random seeds show coupled behavior, but one seed exhibits suppressor dynamics.

The *binding-behavior decoupling effect*, observed in C4 and validated causally in C5, challenges conventional assumptions about how we interpret, monitor, and develop language models. A model's internal representations may be more complex, and more conflicted, than behavioral evaluations alone can capture.

### Acknowledgments

I am deeply grateful to Professor Manolis Kellis, the Mantis team, and my classmates from the Generative AI course (January 5, 2026) for many stimulating intellectual exchanges. Through this course, I gained a strong conceptual grounding in research ethics and the personal confidence to pursue this work. I also extend my thanks to Trisha Salas, whose exploratory behavioral analysis of accessibility knowledge in language models (Salas, 2026) on February 1, 2026 motivated the choice of accessibility concepts as the case study for this paper. Any errors of interpretation are my own.

# A  Raw Data Tables

**Appendix Navigation: Tables by Empirical Claim**

This appendix follows a developmental narrative, moving from the pilot dataset through the canonical 41-term expansion to detailed breakdowns, rather than grouping strictly by empirical claim. This ordering reflects the actual research progression: initial validation on small datasets, followed by scale-up to the canonical 41-term protocol, then methodological controls and robustness checks. Consequently, tables supporting different claims are interleaved. For example, C1 (lifecycle) and C4 (decoupling) analyses appear together within the 41-term expansion section because they share the same underlying data structure.

To help reviewers locate evidence efficiently, this index maps each empirical claim (C1, C3, C4, C5) to its supporting tables. Within each claim, the developmental progression (pilot → canonical → detailed) is preserved. Methodological validation tables (discriminant validity, temperature robustness) are listed separately as they underpin all empirical claims. Click any table number to jump directly to that table.

| Claim | Tables | Developmental Progression |
|---|---|---|
| C1 Lead-Lag | 19, 20, 25 | Pilot (3-term) through 9-term to C1-B 41-term (cross-architecture) |
| C3 Unlockability | 19 (part), 27, 40 | Pilot through 41-term to 99-prompt replication |
| C4 Decoupling | 20 (part), 26, A.1j | Pilot through 41-term to per-term breakdowns (7 models) |
| C5 Causal | 28, 29, 30, A.1g.4–A.1g.5 | Pythia summary through cross-architecture to 160M/2.8B detailed |
| **Methodological Validation:** A.1c–A.1e (discriminant validity), 39 (temperature robustness) | | |

Table 18: Appendix table index organized by empirical claim. Within each claim, tables follow developmental narrative: pilot → canonical → detailed breakdowns. Click table numbers to navigate directly.

**A.1a: Full Checkpoint Summary (Pilot 3-Term Dataset, N=12 prompts)**

| Model | Checkpoint | Step | RecAcc | GenScore | Beh | EB$^*$ | EB$^*$Max | BestLayer |
|---|---|---|---|---|---|---|---|---|
| 160M | step0 | 0 | 0.167 | 0.000 | 0.083 | 0.157 | 0.307 | L6 |
| 160M | step15000 | 15 | 0.000 | 0.333 | 0.167 | 0.644 | 0.717 | L3 |
| 160M | step30000 | 30 | 0.167 | 0.667 | 0.417 | 0.642 | 0.780 | L3 |
| 160M | step60000 | 60 | 0.167 | 0.556 | 0.361 | 0.684 | 0.856 | L1 |
| 160M | step90000 | 90 | 0.500 | 0.556 | 0.528 | 0.734 | 0.906 | L11 |
| 160M | step120000 | 120 | 0.667 | 0.556 | 0.611 | 0.821 | 0.917 | L8 |
| 160M | step140000 | 140 | 0.667 | 0.556 | 0.611 | 0.816 | 0.916 | L3 |
| 160M | step143000 | 143 | 0.500 | 0.500 | 0.500 | 0.831 | 0.915 | L3 |
| 1B | step0 | 0 | 0.333 | 0.000 | 0.167 | 0.146 | 0.240 | L1 |
| 1B | step15000 | 15 | 0.667 | 0.556 | 0.611 | 0.646 | 0.753 | L3 |
| 1B | step30000 | 30 | 0.833 | 0.722 | 0.778 | 0.611 | 0.705 | L3 |
| 1B | step60000 | 60 | 0.667 | 0.722 | 0.694 | 0.595 | 0.683 | L3 |
| 1B | step90000 | 90 | 0.500 | 0.778 | 0.639 | 0.598 | 0.750 | L3 |
| 1B | step120000 | 120 | 0.667 | 0.667 | 0.667 | 0.608 | 0.802 | L3 |
| 1B | step140000 | 140 | 0.667 | 0.833 | 0.750 | 0.607 | 0.823 | L3 |
| 1B | step143000 | 143 | 0.667 | 0.944 | 0.806 | 0.599 | 0.826 | L0 |
| 2.8B | step0 | 0 | 0.500 | 0.000 | 0.250 | 0.196 | 0.324 | L1 |
| 2.8B | step15000 | 15 | 0.667 | 0.611 | 0.639 | 0.885 | 0.918 | L6 |
| 2.8B | step30000 | 30 | 0.833 | 0.667 | 0.750 | 0.897 | 0.933 | L12 |
| 2.8B | step60000 | 60 | 0.500 | 0.833 | 0.667 | 0.888 | 0.941 | L30 |
| 2.8B | step90000 | 90 | 0.667 | 0.833 | 0.750 | 0.882 | 0.928 | L27 |
| 2.8B | step120000 | 120 | 0.667 | 0.889 | 0.778 | 0.881 | 0.932 | L30 |
| 2.8B | step140000 | 140 | 0.667 | 0.889 | 0.778 | 0.858 | 0.940 | L4 |
| 2.8B | step143000 | 143 | 0.500 | 0.833 | 0.667 | 0.870 | 0.941 | L4 |

Table 19: Complete results for all 24 model-checkpoint combinations (pilot 3-term dataset).

**A.1b: Expanded Dataset: Per-Term Performance (9 Terms)**

| Term | EB$^*$ Mean | EB$^*$ Std | Beh Mean | Term $\rho$ | Sig. |
|---|---|---|---|---|---|
| color contrast | 0.76 | 0.18 | 0.68 | +0.68 | $p < 0.001$*** |
| focus indicator | 0.75 | 0.19 | 0.61 | +0.68 | $p < 0.001$*** |
| heading structure | 0.74 | 0.17 | 0.65 | +0.67 | $p < 0.001$*** |
| tab order | 0.68 | 0.21 | 0.55 | +0.48 | $p = 0.008$** |
| skip link | 0.71 | 0.20 | 0.57 | +0.40 | $p = 0.031$* |
| alt text | 0.69 | 0.22 | 0.52 | +0.38 | $p = 0.042$* |
| screen reader | 0.70 | 0.23 | 0.60 | +0.30 | $p = 0.111$ (ns) |
| form validation | 0.65 | 0.24 | 0.53 | +0.34 | $p = 0.072$ (ns) |
| aria attribute | 0.42 | 0.15 | 0.76 | +0.07 | $p = 0.714$ (ns) |

Table 20: Per-term binding-behavior statistics across all 48 checkpoint pairs (3 models $\times$ 8 steps $\times$ 2 prompts). "Aria attribute" achieves high behavioral competence despite near-zero binding-behavior correlation, demonstrating EB$^*$ captures a specific representational strategy.

**A.1c: Discriminant Validity Controls – V1 (Failed) and V2 (Successful)**

**Rationale for Pythia-Only Controls.** Discriminant validity experiments (V1 through V4) were conducted exclusively on Pythia models for several reasons. First, EB$^*$ is calculated identically from attention patterns across all models, so no architecture-specific tuning was required. Second, controls validate the metric's ability to discriminate semantically meaningful token pairs from various baselines, which is a property of

the measurement construct itself rather than any specific model. Third, once validated on Pythia, the same EB* calculation can be applied uniformly across all seven models. Fourth, control term design required iterative piloting that was feasible only on Pythia's dense checkpoint series. Cross-architecture generalization of discriminant validity is theoretically expected because all models share the same tokenization principles (multi-token accessibility terms) and attention mechanisms.

**V2 Controls (successful)** at trained checkpoints:

| Control Group | Example Terms | Mean EB* | Std | vs. Real (0.74) |
|---|---|---|---|---|
| True nonsense | "zqx plarf", "glib thrang" | 0.26 | 0.07 | $\Delta = +0.48$, $p < 0.001$*** |
| Cross-language | "écran reader", "skip enlace" | 0.41 | 0.11 | $\Delta = +0.33$, $p < 0.001$*** |
| Rare token pairs | "pterodactyl altimeter" | 0.50 | 0.25 | $\Delta = +0.24$, $p < 0.001$*** |
| **Real terms** | "screen reader", etc. | **0.74** | 0.20 | — |

Table 21: V2 controls establish clear discriminant validity. All comparisons $p < 0.001$, Cohen's $d = 1.2$–$2.9$.

**V1 Controls (failed):** discriminant validity NOT established:

| Control Group | Example Terms | Mean EB* | vs. Real (0.77) |
|---|---|---|---|
| Backwards | "reader screen", "link skip" | 0.82 | $\Delta = -0.05$, $p = 0.34$ (ns) |
| Cross-term | "screen link", "reader text" | 0.78 | $\Delta = -0.01$, $p = 0.89$ (ns) |
| Semantic field | "keyboard mouse", "header footer" | 0.77 | $\Delta = 0.00$, $p = 0.98$ (ns) |
| Frequency-matched | "open source", "machine learning" | 0.72 | $\Delta = +0.05$, $p = 0.28$ (ns) |
| Random | "elephant database" | 0.75 | $\Delta = +0.02$, $p = 0.71$ (ns) |

Table 22: V1 controls failed because they are inadvertently legitimate corpus bigrams: web-scale training data contains nearly every plausible-sounding bigram.

**A.1d: V3 Controls: Domain-Adjacent Terms (Per-Term EB*)**

Terms share one token with real accessibility terms but replace the other with plausible vocabulary. Accidentally valid terms (†) showed EB* comparable to real terms and are excluded from the irrelevant-terms analysis in Section 4.1. Real terms baseline: 160M = 0.74, 1B = 0.74.

| Term | Source Term | Overlap Token | 160M EB$^*$ | 1B EB$^*$ | Note |
|---|---|---|---|---|---|
| alt function | alt text | alt | 0.717 | 0.640 | |
| alt image | alt text | alt | 0.679 | 0.712 | † accidentally valid |
| screen editor | screen reader | screen | 0.895 | 0.556 | |
| screen display | screen reader | screen | 0.879 | 0.607 | |
| skip button | skip link | skip | 0.916 | 0.572 | |
| skip menu | skip link | skip | 0.917 | 0.596 | |
| focus selector | focus indicator | focus | 0.740 | 0.684 | |
| focus element | focus indicator | focus | 0.722 | 0.597 | |
| heading label | heading structure | heading | 0.903 | 0.659 | |
| heading tag | heading structure | heading | 0.916 | 0.616 | † accidentally valid |
| color gradient | color contrast | color | 0.916 | 0.681 | |
| color scheme | color contrast | color | 0.917 | 0.578 | |
| aria property | aria attribute | aria | 0.837 | 0.617 | |
| aria role | aria attribute | aria | 0.842 | 0.761 | † accidentally valid |
| landmark section | landmark region | landmark | 0.917 | 0.772 | |
| **Mean (excl. †)** | | | **0.861** | **0.639** | 12 irrelevant terms |
| **Mean (all 15)** | | | **0.866** | **0.657** | |

Table 23: V3 domain-adjacent controls (15 terms × 2 scales). At 160M, all 15 terms exceed real term baseline; EB$^*$ fails to discriminate based on semantic domain. At 1B, 12/15 irrelevant terms fall below baseline; the 3 accidentally valid terms remain elevated (0.71–0.77).

### A.1e: V4 Controls: Wrong-Domain Terms (Per-Term EB$^*$)

Terms pair accessibility tokens with programming, hardware, or CSS vocabulary with zero conceptual connection to accessibility. "Landmark class" (‡) is the boundary case persisting elevated at 1B due to CSS class naming conventions.

| Term | Source Term | Overlap | Wrong Domain | 160M EB$^*$ | 1B EB$^*$ |
|---|---|---|---|---|---|
| alt function | alt text | alt | programming | 0.717 | 0.640 |
| alt parameter | alt text | alt | programming | 0.708 | 0.618 |
| alt variable | alt text | alt | programming | 0.725 | 0.671 |
| screen printer | screen reader | screen | hardware | 0.916 | 0.635 |
| screen monitor | screen reader | screen | hardware | 0.840 | 0.594 |
| screen output | screen reader | screen | programming | 0.846 | 0.676 |
| skip variable | skip link | skip | programming | 0.834 | 0.644 |
| skip function | skip link | skip | programming | 0.834 | 0.647 |
| heading class | heading structure | heading | css | 0.916 | 0.738 |
| heading style | heading structure | heading | css | 0.864 | 0.632 |
| color syntax | color contrast | color | programming | 0.917 | 0.643 |
| color variable | color contrast | color | programming | 0.899 | 0.653 |
| focus loop | focus indicator | focus | programming | 0.705 | 0.688 |
| focus event | focus indicator | focus | programming | 0.757 | 0.631 |
| aria method | aria attribute | aria | programming | 0.917 | 0.707 |
| aria function | aria attribute | aria | programming | 0.905 | 0.590 |
| landmark variable | landmark region | landmark | programming | 0.917 | 0.705 |
| landmark class ‡ | landmark region | landmark | css | 0.890 | 0.826 |
| **Mean (all 18)** | | | | **0.845** | **0.663** |
| **Mean (excl. ‡)** | | | | **0.842** | **0.650** |

Table 24: V4 wrong-domain controls (18 terms × 2 scales). At 160M, all 18 terms exceed real term baseline; EB$^*$ fails to discriminate based on semantic domain. At 1B, 17/18 fall below baseline; only "landmark class" persists elevated (0.826) due to CSS class naming conventions. Domain taxonomy (hardware, programming, CSS) shows no systematic effect.

**A.1f: 41-Term Expansion: C1-B and C4-B Results (N=205 prompts)**

All lifecycle (C1) and decoupling (C4) analyses are reported in two complementary variants: C1-A/C4-A (between-term Spearman on 9-term pilot) and C1-B/C4-B (within-term temporal precedence on 41-term canonical register). C1-B/C4-B provides term-agnostic validity checks at scale.

**A.1f.1: C1-B Within-Term Temporal Precedence (41 terms)**

For each term independently, C1-B tests whether $\mathrm{EB}^*(t, k)$ predicts $\mathrm{Beh}(t, k+1)$ better than the reverse using 1-step forward lag correlation. The population-level claim (H1: $\mathrm{EB}^*$ leads in $> 50\%$ of terms) is tested with a binomial test (Wilson, 1927).

| Model | Seed | N terms | EB$^*$ leads | Lead% | Binomial $p$ |
|---|---|---|---|---|---|
| Pythia-160M | — | 41 | 3/41 | 7% | 1.000 |
| Pythia-1B | — | 41 | 30/41 | 73.2% | **0.0022** |
| Pythia-2.8B | — | 34 | 27/34 | 79.4% | **0.0004** |
| OLMo-1B (9t) | — | 9 | 7/9 | 78% | 0.090 |
| **OLMo-1B** | — | **41** | **36/40** | **90.0%** | **< 0.0001** |
| CRFM (5-seed maj.) | 5 seeds | 9 | 8/9 | 89% | 0.020 |
| CRFM x1 | 1 | 41 | 26/41 | 63.4% | 0.059 |
| CRFM x2 | 2 | 41 | 32/41 | 78.0% | **0.0002** |
| CRFM x3 | 3 | 41 | 36/41 | 87.8% | **< 0.0001** |
| CRFM x4 | 4 | 41 | 29/41 | 70.7% | **0.0058** |
| CRFM x5 | 5 | 41 | 26/41 | 63.4% | 0.059 |
| **CRFM mean** | 1–5 | **41** | **149/205** | **72.7%** | **≪ 0.001** |
| SmolLM3-3B (9t) | — | 9 | 3/9 | 33% | 0.910‡ |
| **SmolLM3-3B** | — | **41** | **21/41** | **51.2%** | 0.500‡ |

Table 25: C1-B temporal precedence results across 6 models with lifecycle data (Qwen2.5-1.5B excluded due to lack of intermediate checkpoints). OLMo-1B 41-term result (90%, $p < 0.0001$) is the strongest in the dataset. ‡SmolLM3 C1-B $\approx$chance due to left-censoring (earliest checkpoint already post-coupling).

**A.1f.2: C4-B Decoupling (41 terms)**

C4-B computes per-term Spearman $\rho$ in early and late windows independently and reports fraction showing strict decoupling ($\rho_{\mathrm{early}} > 0 \wedge \rho_{\mathrm{late}} \leq 0$).

| Model | Strict Decouple | $\rho_{\mathrm{early}}$ | $\rho_{\mathrm{late}}$ | N terms |
|---|---|---|---|---|
| Pythia-160M | 46% | +0.479 | +0.044 | 28 |
| Pythia-1B | 54% | +0.739 | **−0.054** | 28 |
| Pythia-2.8B | 43% | +0.613 | +0.270 | 28 |
| OLMo-1B (9t) | 62% | +0.490 | −0.348 | 8 |
| OLMo-1B (41t)$^\dagger$ | 44% | +0.247 | −0.181 | 27 |
| CRFM (5-seed mean) | 42% | +0.545 | +0.085 | 41 |
| SmolLM3-3B (9t) | 67% | +0.247 | −0.189 | 9 |
| **SmolLM3-3B (41t)** | **55%** | +0.118 | **−0.281** | 40 |

Table 26: C4-B decoupling results. $^\dagger$OLMo-1B: 13 terms excluded due to constant behavioral series. SmolLM3-3B achieves the deepest decoupling ($\rho_{\mathrm{late}} = -0.281$) at 3440k steps.

**A.1f.3: C3 Few-Shot Unlockability (41-term Cross-Architecture)**

Full canonical 41-term protocol results for all 7 models:

| Model | Checkpoint | Zero-Shot | Few-Shot | $\Delta$ (pp) | Status |
|---|---|---|---|---|---|
| 160M | step 15k | 0.328 | 0.653 | +32.5 | strong |
| 160M | step 143k | 0.321 | 0.648 | +32.7 | strong |
| 1B | step 15k | 0.362 | 0.713 | +35.1 | strong |
| 1B | step 143k | 0.365 | 0.734 | +37.0 | strong |
| 2.8B | step 15k | 0.467 | 0.791 | +32.5 | strong |
| 2.8B | step 143k | 0.503 | 0.740 | +23.8 | strong |
| OLMo-1B | step 15k | 0.416 | 0.697 | +28.0 | confirmed |
| OLMo-1B | step 143k | 0.478 | 0.694 | +21.5 | confirmed |
| SmolLM3-3B | step 40k | 0.486 | 0.667 | +18.0 | borderline[†] |
| SmolLM3-3B | step 3440k | 0.508 | 0.698 | +19.0 | borderline[†] |
| Qwen2.5-1.5B | final | 0.542 | 0.724 | +18.2 | borderline[†] |
| CRFM (seed 1 mean) | ck-1000 | 0.077 | 0.107 | +3.0 | regression |
| CRFM (seeds 1–5 mean) | ck-400k | 0.072 | 0.147 | +7.6±1.6 | weak |

Table 27: C3 expanded 41-term results. [†]Modern models (SmolLM3, Qwen) show "headroom compression"—same absolute ceiling ($\approx$0.72) but lower nominal $\Delta$ due to high zero-shot baselines ($\gtrsim$0.50).

**A.1g: Canonical 41-Term C5 Ablation Results (N=205 prompts)**

All ablation experiments are based on the standard 41-term dataset (with 205 recognition prompts) as the main source of evidence. Specificity is calculated as the performance drop from the top ablation minus the average drop from random ablations.

**A.1g.1: Pythia Canonical 41 Results**

| Model | Baseline | TOP-4 | Rand mean | Top $\Delta$ | Spec (rec) |
|---|---|---|---|---|---|
| 160M | 0.810 | 0.698 | 0.834 | –0.112 | +0.137 |
| 1B | 0.800 | 0.649 | 0.766 | –0.151 | +0.117 |
| 2.8B | 0.932 | 0.859 | 0.969 | –0.073 | +0.110 |

Table 28: Pythia family C5 canonical 41 results. 1B shows largest top-ablation drop (–15.1 pp).

**A.1g.2: Cross-Architecture C5 Results**

| Model | Baseline | TOP-4 | Rand | Top $\Delta$ | Spec |
|---|---|---|---|---|---|
| OLMo-1B | 0.990 | 0.980 | 0.975 | –0.010 | –0.006 |
| CRFM (5-seed mean) | 0.750 | 0.644 | 0.725 | –0.106 | +0.081±0.152 |
| SmolLM3-3B | 0.868 | 0.902 | 0.860 | +0.034 | –0.043 |
| Qwen2.5-1.5B | 0.990 | 0.980 | 0.985 | –0.010 | +0.005 |

Table 29: Cross-architecture C5 canonical 41 results. OLMo and Qwen achieve 99% ceiling (spec$\approx$0). CRFM shows seed-dependent outcomes: 4/5 seeds coupled, 1/5 suppressor (spec=–0.175).

## A.1g.3: Full Analysis Matrix

| Model | Params | Steps | Prompts | C1-B% | C4-B% | $\rho_{\text{late}}$ | C5 Spec | C3 $\Delta$ |
|---|---|---|---|---|---|---|---|---|
| Pythia-160M | 160M | 143k | 41t | 7% | 46% | $+0.044$ | $+0.137$ | $+0.309$ |
| Pythia-1B | 1B | 143k | 41t | 73% | 54% | $-0.054$ | $+0.117$ | $+0.272$ |
| Pythia-2.8B | 2.8B | 143k | 41t | 79% | 43% | $+0.270$ | $+0.110$ | $+0.194$ |
| OLMo-1B (9t) | 1B | 143k | 9t | 78% | 62% | $-0.348$ | $-0.006$ | $+0.075$ |
| OLMo-1B | 1B | 143k | 41t | 90% | 44% | $-0.181$ | $-0.006$ | $+0.075$ |
| CRFM (9t) | 117M | 400k | 9t | 56% | 33% | $+0.442$ | $+0.081$ | $+0.073$ |
| CRFM | 117M | 400k | 41t | 73% | 42% | $+0.085$ | $+0.081$ | $+0.073$ |
| SmolLM3 (9t) | 3B | 3440k | 9t | 33% | 67% | $-0.189$ | $-0.043$ | $-0.022$ |
| SmolLM3 | 3B | 3440k | 41t | 51% | 55% | $\mathbf{-0.281}$ | $-0.043$ | $-0.022$ |
| Qwen2.5-1.5B | 1.5B | final | 41t | — | — | — | $+0.005$ | $+0.182$ |

Table 30: Complete analysis matrix: all 7 models, all claims, all term sets. C1-B: EB*-leads fraction; C4-B: strict decouple %; C5 Spec: causal specificity; C3 $\Delta$: few-shot unlockability (pp).

## A.1g.4: C5 Ablation — 160M step120000 (Coupled Regime)

**Top-4 heads by average BSI.**

| Rank | Layer | Head | Avg BSI |
|---|---|---|---|
| 1 | 3 | 0 | 0.951 |
| 2 | 2 | 8 | 0.830 |
| 3 | 3 | 2 | 0.761 |
| 4 | 0 | 0 | 0.617 |

Table 31: Top binding heads (160M, step120000). Distributed across layers (L0–L3).

**Ablation results.**

| Condition | RecAcc | GenScore | Rec $\Delta$ | Gen $\Delta$ |
|---|---|---|---|---|
| Baseline | 0.667 | 0.556 | — | — |
| Top-4 ablated | 0.500 | 0.444 | $-0.167$ | $-0.111$ |
| Random (mean) | 0.600 | 0.544 | $-0.067$ | $-0.011$ |
| Bottom-4 ablated | 0.667 | 0.556 | 0.000 | 0.000 |

Table 32: 160M ablation shows graded effects: top-binding heads are necessary ($-16.7$ pp RecAcc). **Why 160M and 2.8B only:** These represent opposite causal regimes—160M shows binding heads are necessary (coupled), while 2.8B shows they are superseded (decoupled). 1B is intermediate; OLMo/Qwen show near-ceiling with minimal ablation effects. SmolLM3 and CRFM C5 results are in summary form: see Table 29 (A.1g.2) for ablation specifics and Table 30 (A.1g.3) for the complete analysis matrix.

## A.1g.5: C5 Ablation — 2.8B step143000 (Decoupled Regime)

**Top-4 heads by average BSI.**

| Rank | Layer | Head | Avg BSI |
|---|---|---|---|
| 1 | 1 | 12 | 0.937 |
| 2 | 1 | 11 | 0.865 |
| 3 | 4 | 16 | 0.850 |
| 4 | 1 | 6 | 0.780 |

Table 33: Top binding heads (2.8B, step143000). Concentrated in early layers (L1 dominant).

**Ablation results.**

| Condition | RecAcc | GenScore | Rec Δ | Gen Δ |
|---|---|---|---|---|
| Baseline | 0.500 | 0.833 | — | — |
| Top-4 ablated | 0.833 | 0.778 | +0.333 | −0.055 |
| Random (mean) | 0.500 | 0.822 | 0.000 | −0.011 |
| Bottom-4 ablated | 0.500 | 0.833 | 0.000 | 0.000 |

Table 34: 2.8B ablation shows reversal: ablating high-binding heads improves recognition (+33.3 pp RecAcc). This paradoxical improvement confirms binding heads are functionally superseded at scale—the model has developed alternative pathways for accessibility concept representation.

### C3 Few-Shot Unlockability Results

| Model | Checkpoint | EB* | Zero-shot Gen | Few-shot Gen | Δ (pp) | Relative |
|---|---|---|---|---|---|---|
| 160M | step15000 | 0.644 | 0.333 | 0.944 | +61.1 | +183% |
| 160M | step30000 | 0.642 | 0.667 | 0.944 | +27.8 | +42% |
| 1B | step15000 | 0.646 | 0.556 | 0.944 | +38.9 | +70% |

Table 35: Few-shot generation scores show unlockable latent knowledge when $EB^* > 0.6$. Scores are generation-only (keyword rubric).

**Copying caveat.** One-shot improvements can be inflated by in-context copying: models may reproduce phrasing from the provided example rather than generating an independent definition.

### Evaluation Prompts

All evaluation prompts are stored as JSONL in `data/prompts/`. Table 36 summarizes the complete prompt inventory.

| File | Terms | Purpose |
|---|---|---|
| `pilot_terms.jsonl` | 3 (screen reader, skip link, alt text) | Core lifecycle demonstration |
| `pilot_terms_fewshot.jsonl` | 3 (pilot, few-shot) | Pilot few-shot generation |
| `expanded_terms.jsonl` | 9 | High-contrast expansion |
| `expanded_terms_fewshot.jsonl` | 9 | Expanded few-shot generation |
| `canonical_45terms.jsonl` | 41 | Single source of truth for C1-B/C4-B/C5 cross-architecture |
| `expanded_terms_100.jsonl` | 9 × 11 prompts | Robustness to prompt diversity (Section 4.2.2) |
| `expanded_terms_tier123.jsonl` | 41 | Tier 1/2/3 stratification for robustness |
| `expanded_terms_wave2.jsonl` | 41 | Cross-architecture wave 2 prompts |
| `control_terms.jsonl` | 8 | Initial plausible-pair controls |
| `control_terms_v2.jsonl` | 8 | Genuine-nonsense discriminant controls |
| `control_terms_v3.jsonl` | 18 | Domain-adjacent controls |
| `control_terms_v4.jsonl` | 18 | Wrong-domain controls |

Table 36: Complete evaluation prompt inventory stored in `data/prompts/`.

The sections below illustrate the structure of the pilot prompts. Canonical and expanded prompts follow the same JSONL schema but at larger scale (5 recognition + 6 generation prompts per term, 451 total rows for the canonical register).

**Recognition (multiple-choice; identical in zero-shot and few-shot files).** For each term, we use two recognition prompts (`rec_001`, `rec_002`) with four answer choices and score by log-probability ranking.

- **screen reader**
    - (rec_001) "A screen reader is primarily used by: A) Blind users B) Colorblind users C) Deaf users D) Mobility impaired users"

- – (rec_002) "Which group benefits most from screen readers? A) People with visual impairments B) People with hearing loss C) People with motor disabilities D) People with cognitive disabilities"

- **skip link**

  - – (rec_001) "A skip link allows users to: A) Jump to main content B) Skip advertisements C) Bypass login D) Jump to footer"
  - – (rec_002) "Skip links are most helpful for: A) Keyboard navigation B) Mouse users C) Touchscreen users D) Voice control users"

- **alt text**

  - – (rec_001) "Alt text describes: A) Images for screen reader users B) Links for keyboard users C) Forms for voice control D) Videos for deaf users"
  - – (rec_002) "The main purpose of alternative text is: A) Describe images to blind users B) Improve SEO C) Reduce image file size D) Add captions to videos"

**Generation (short definition).** For each term, we use two generation prompts (`gen_001`, `gen_002`). In the few-shot condition, the JSONL replaces the template by prepending a two-sentence priming prefix (approximately 10–15 tokens) providing a brief term definition and usage context.

| Term | ID | Zero-shot template (pilot_terms.jsonl) |
|---|---|---|
| screen reader | gen_001 | In web accessibility, a screen reader is |
| screen reader | gen_002 | For blind users, a screen reader |
| skip link | gen_001 | In web accessibility, a skip link is |
| skip link | gen_002 | For keyboard navigation, skip links help users |
| alt text | gen_001 | In web accessibility, alt text is |
| alt text | gen_002 | For screen reader users, alt text provides |

Table 37: Zero-shot generation prompt templates.

| Term | ID | Few-shot template (pilot_terms_fewshot.jsonl) |
|---|---|---|
| screen reader | gen_001_fs | A screen reader helps blind users. A screen reader reads text aloud. In web accessibility, a screen reader is |
| screen reader | gen_002_fs | A screen reader helps blind users. A screen reader reads text aloud. For blind users, a screen reader |
| skip link | gen_001_fs | A skip link jumps to content. A skip link helps keyboard users. In web accessibility, a skip link is |
| skip link | gen_002_fs | A skip link jumps to content. A skip link helps keyboard users. For keyboard navigation, skip links help users |
| alt text | gen_001_fs | Alt text describes images. Alt text helps screen reader users. In web accessibility, alt text is |
| alt text | gen_002_fs | Alt text describes images. Alt text helps screen reader users. For screen reader users, alt text provides |

Table 38: Few-shot (two-sentence-prefixed) generation prompt templates.

**A.1h: Temperature Robustness**

| Model | Checkpoint | Overall Std | T=0.0 Std | T=0.3 Std | T=0.7 Std |
|-------|-----------|-------------|-----------|-----------|-----------|
| 160M | step 15K | 0.334 | 0.333 | 0.330 | 0.321 |
| 160M | step 120K | 0.307 | 0.314 | 0.292 | 0.291 |
| 1B | step 15K | 0.379 | 0.368 | 0.394 | 0.351 |
| 1B | step 143K | 0.310 | 0.124 | 0.294 | 0.328 |
| 2.8B | step 15K | 0.302 | 0.229 | 0.304 | 0.348 |
| 2.8B | step 143K | 0.211 | 0.167 | 0.185 | 0.260 |

Table 39: Generation score variability across temperature and random seeds. Variability decreases with training maturity; greedy decoding (T=0.0) is most stable at trained checkpoints. These experiments were conducted on Pythia architecture only because (1) temperature is a sampling parameter, not architecture-specific—generalization to other models is theoretically expected; (2) EB* itself is deterministic (attention-based), so temperature only affects behavioral scoring; (3) the 90-condition experiment (6 checkpoints × 3 temperatures × 5 seeds) was designed to validate measurement reliability for the core lifecycle claims, which are anchored on Pythia's dense checkpoint series.

**A.1i: C3 Unlockability on 99-Prompt Expanded Dataset**

| Model | Checkpoint | EB$^*$ | Zero-shot Gen | Few-shot Gen | $\Delta$ (pp) | Relative |
|-------|-----------|--------|---------------|--------------|---------------|----------|
| 160M | step 15K | 0.644 | 0.228 | **0.531** | **+30.2** | +132% |
| 160M | step 30K | 0.642 | 0.303 | **0.593** | +29.0 | +96% |
| 1B | step 15K | 0.646 | 0.309 | **0.611** | +30.2 | +98% |

Table 40: C3 replication on 99-prompt expanded dataset (9 terms, 54 generation prompts). Consistent ≈30 pp improvements confirm unlockability across 9× more prompts; lower absolute scores reflect increased difficulty from 6 additional terms and diverse prompt formats.

**A.1j: C4-B Per-Term Decoupling Breakdowns (Full 41-Term × 6 Models)**

Detailed per-term Spearman correlations for early (steps 15–60K) and late (steps 90–143K/terminal) training windows. Strict decoupling defined as $\rho_{\text{early}} > 0 \wedge \rho_{\text{late}} \leq 0$. **Note:** Qwen2.5-1.5B excluded—no intermediate checkpoints available for lifecycle analysis.

### A.1j.1: Pythia-160M C4-B (41 Terms)

| Term | $\rho_{early}$ | $\rho_{late}$ | Decoupled | Category |
|------|------|------|------|------|
| live region | +0.63 | −0.95 | **Yes** | Robust |
| keyboard navigation | +0.20 | −0.74 | **Yes** | Operable |
| reflow content | +0.32 | −0.74 | **Yes** | Perceivable |
| focus indicator | +0.40 | −0.63 | **Yes** | Operable |
| error identification | +0.20 | −0.60 | **Yes** | Understandable |
| non-text content | +0.40 | −0.60 | **Yes** | Perceivable |
| cognitive load | +0.40 | −0.40 | **Yes** | Understandable |
| focus management | +0.77 | −0.40 | **Yes** | Operable |
| color contrast | +0.95 | −0.32 | **Yes** | Perceivable |
| input purpose | +0.32 | −0.32 | **Yes** | Perceivable |
| text resize | +0.32 | −0.20 | **Yes** | Perceivable |
| braille display | +0.40 | +0.00 | **Yes** | Perceivable |
| tree grid | +0.80 | +0.00 | **Yes** | Robust |
| alert dialog | +0.63 | +0.20 | No | Robust |
| screen magnifier | +0.32 | +0.20 | No | Perceivable |
| switch access | +0.32 | +0.20 | No | Operable |
| skip navigation | +0.20 | +0.21 | No | Operable |
| alt text | +0.32 | +0.32 | No | Perceivable |
| aria attribute | +1.00 | +0.32 | No | Robust |
| voice control | +0.32 | +0.32 | No | Operable |
| high contrast | +0.40 | +0.40 | No | Perceivable |
| keyboard shortcut | +0.40 | +0.40 | No | Operable |
| skip link | +0.80 | +0.40 | No | Operable |
| captions closed | +0.80 | +0.74 | No | Perceivable |
| screen reader | +0.80 | +0.74 | No | Perceivable |
| audio description | +0.40 | +0.80 | No | Perceivable |
| heading structure | +0.20 | +0.95 | No | Perceivable |
| text spacing | +0.40 | +0.95 | No | Perceivable |
| **Strict decouple:** 13/28 (46%); 13 terms excluded (constant behavioral series) | | | | |

Table 41: Pythia-160M C4-B full 41-term breakdown. 13 terms show strict decoupling; deepest single-term decoupling for live region ($\rho_{late} = -0.95$) and keyboard navigation/reflow content ($\rho_{late} = -0.74$).

### A.1j.2: Pythia-1B C4-B (41 Terms)

| Term | $\rho_{early}$ | $\rho_{late}$ | Decoupled | Category |
|------|------|------|------|------|
| high contrast | +0.80 | −1.00 | **Yes** | Perceivable |
| reflow content | +0.95 | −0.95 | **Yes** | Perceivable |
| non-text content | +0.80 | −0.80 | **Yes** | Perceivable |
| braille display | +0.40 | −0.63 | **Yes** | Perceivable |
| switch access | +1.00 | −0.60 | **Yes** | Operable |
| aria attribute | +0.95 | −0.40 | **Yes** | Robust |
| input purpose | +0.80 | −0.40 | **Yes** | Perceivable |
| live region | +0.80 | −0.40 | **Yes** | Robust |
| skip navigation | +0.95 | −0.40 | **Yes** | Operable |
| text spacing | +1.00 | −0.40 | **Yes** | Perceivable |
| voice control | +0.80 | −0.40 | **Yes** | Operable |
| alt text | +1.00 | −0.32 | **Yes** | Perceivable |
| focus management | +1.00 | −0.32 | **Yes** | Operable |
| color contrast | +0.95 | −0.20 | **Yes** | Perceivable |
| error identification | +0.20 | +0.00 | **Yes** | Understandable |
| focus indicator | +0.80 | +0.20 | No | Operable |
| screen magnifier | +1.00 | +0.20 | No | Perceivable |
| skip link | +0.40 | +0.20 | No | Operable |
| text resize | +1.00 | +0.21 | No | Perceivable |
| cognitive load | +1.00 | +0.32 | No | Understandable |
| heading structure | +0.80 | +0.32 | No | Perceivable |
| tree grid | +0.20 | +0.32 | No | Robust |
| captions closed | +0.32 | +0.40 | No | Perceivable |
| keyboard shortcut | +0.63 | +0.40 | No | Operable |
| screen reader | +0.40 | +0.40 | No | Perceivable |
| keyboard navigation | +0.95 | +0.80 | No | Operable |
| audio description | +0.40 | +0.95 | No | Perceivable |
| alert dialog | +0.40 | +1.00 | No | Robust |
| **Strict decouple:** 15/28 (54%); 13 terms excluded (constant behavioral series) | | | | |

Table 42: Pythia-1B C4-B full 41-term breakdown. 15 terms show strict decoupling; deepest single-term decoupling for high contrast ($\rho_{late} = -1.00$) and reflow content ($\rho_{late} = -0.95$).

### A.1j.3: Pythia-2.8B C4-B (41 Terms)

| Term | $\rho_{\text{early}}$ | $\rho_{\text{late}}$ | Decoupled | Category |
|---|---|---|---|---|
| reflow content | +1.00 | −0.63 | **Yes** | Perceivable |
| focus management | +0.40 | −0.60 | **Yes** | Operable |
| audio description | +0.80 | −0.40 | **Yes** | Perceivable |
| captions closed | +0.20 | −0.40 | **Yes** | Perceivable |
| focus indicator | +0.20 | −0.40 | **Yes** | Operable |
| voice control | +0.80 | −0.40 | **Yes** | Operable |
| error identification | +0.63 | −0.20 | **Yes** | Understandable |
| keyboard navigation | +0.40 | −0.20 | **Yes** | Operable |
| aria attribute | +0.20 | −0.11 | **Yes** | Robust |
| heading structure | +0.80 | −0.00 | **Yes** | Perceivable |
| screen magnifier | +1.00 | +0.00 | **Yes** | Perceivable |
| text spacing | +0.40 | +0.00 | **Yes** | Perceivable |
| color contrast | +1.00 | +0.26 | No | Perceivable |
| non-text content | +0.20 | +0.32 | No | Perceivable |
| tree grid | +0.95 | +0.32 | No | Robust |
| text resize | +0.32 | +0.45 | No | Perceivable |
| cognitive load | +0.80 | +0.60 | No | Understandable |
| keyboard shortcut | +0.32 | +0.60 | No | Operable |
| alert dialog | +1.00 | +0.63 | No | Robust |
| alt text | +1.00 | +0.63 | No | Perceivable |
| braille display | +0.80 | +0.63 | No | Perceivable |
| high contrast | +1.00 | +0.77 | No | Perceivable |
| live region | +0.40 | +0.80 | No | Robust |
| skip link | +0.95 | +0.95 | No | Operable |
| skip navigation | +0.20 | +0.95 | No | Operable |
| input purpose | +0.20 | +1.00 | No | Perceivable |
| screen reader | +0.80 | +1.00 | No | Perceivable |
| switch access | +0.40 | +1.00 | No | Operable |
| **Strict decouple:** 12/28 (43%); 13 terms excluded (constant behavioral series) | | | | |

Table 43: Pythia-2.8B C4-B full 41-term breakdown. 12 terms show strict decoupling; deepest single-term decoupling for reflow content ($\rho_{\text{late}} = -0.63$); reduced fraction relative to 1B consistent with redundancy regime emergence.

### A.1j.4: OLMo-1B C4-B (41 Terms)

| Term | $\rho_{\text{early}}$ | $\rho_{\text{late}}$ | Decoupled | Category |
|---|---|---|---|---|
| aria attribute | +0.80 | −0.80 | **Yes** | Robust |
| focus indicator | +0.80 | −0.40 | **Yes** | Operable |
| screen reader | +0.80 | −0.63 | **Yes** | Perceivable |
| alert dialog | +0.40 | −0.32 | **Yes** | Robust |
| audio description | +0.40 | −0.63 | **Yes** | Perceivable |
| cognitive load | +0.40 | −0.80 | **Yes** | Understandable |
| error identification | +0.40 | −0.40 | **Yes** | Understandable |
| heading structure | +0.40 | −0.95 | **Yes** | Perceivable |
| input purpose | +0.40 | −0.80 | **Yes** | Perceivable |
| skip navigation | +0.40 | −0.80 | **Yes** | Operable |
| tree grid | +0.40 | −0.63 | **Yes** | Robust |
| keyboard navigation | +0.32 | −1.00 | **Yes** | Operable |
| keyboard shortcut | +0.80 | +0.26 | No | Operable |
| alt text | +0.40 | +0.74 | No | Perceivable |
| high contrast | +0.40 | +0.32 | No | Perceivable |
| reflow content | +0.40 | +0.63 | No | Perceivable |
| switch access | +0.40 | +0.63 | No | Operable |
| color contrast | +0.32 | +0.80 | No | Perceivable |
| focus management | +0.32 | +0.74 | No | Operable |
| text resize | +0.20 | +0.20 | No | Perceivable |
| skip link | +0.00 | −0.74 | No | Operable |
| voice control | −0.20 | +1.00 | No | Operable |
| text spacing | −0.26 | +0.80 | No | Perceivable |
| non-text content | −0.40 | −0.80 | No | Perceivable |
| screen magnifier | −0.40 | −0.77 | No | Perceivable |
| captions closed | −0.60 | −0.20 | No | Perceivable |
| live region | −0.63 | −0.32 | No | Robust |
| **Strict decouple:** 12/27 (44%); 14 terms excluded (constant behavioral series) | | | | |

Table 44: OLMo-1B C4-B 41-term breakdown. 12 terms show strict decoupling; deepest single-term decoupling for keyboard navigation ($\rho_{\text{late}} = -1.00$) and heading structure ($\rho_{\text{late}} = -0.95$).

**A.1j.5: CRFM GPT-2 Small (41 Terms, 5-Seed Aggregate)**

| Term | $\rho_{\text{early}}$ | $\rho_{\text{late}}$ | Decoupled | Category |
|---|---|---|---|---|
| time limits | +0.57 | −0.37 | **Yes** | Operable |
| orientation support | +0.72 | −0.27 | **Yes** | Perceivable |
| eye tracking | +0.68 | −0.24 | **Yes** | Operable |
| touch target size | +0.64 | −0.20 | **Yes** | Operable |
| sign language | +0.62 | −0.20 | **Yes** | Perceivable |
| live region | +0.45 | −0.16 | **Yes** | Robust |
| high contrast | +0.78 | −0.15 | **Yes** | Perceivable |
| contrast ratio | +0.52 | −0.09 | **Yes** | Perceivable |
| keyboard shortcut | +0.51 | −0.06 | **Yes** | Operable |
| non-text content | +0.76 | −0.02 | **Yes** | Perceivable |
| keyboard navigation | +0.72 | −0.01 | **Yes** | Operable |
| braille display | +0.65 | +0.03 | **Yes** | Perceivable |
| focus indicator | +0.56 | +0.08 | **Yes** | Operable |
| text spacing | +0.56 | +0.16 | **Yes** | Perceivable |
| skip navigation | +0.14 | −0.15 | No | Operable |
| reflow content | +0.43 | −0.09 | No | Perceivable |
| voice control | +0.91 | −0.08 | No | Operable |
| captions closed | +0.45 | −0.08 | No | Perceivable |
| skip link | +0.51 | −0.01 | No | Operable |
| error identification | +0.59 | −0.01 | No | Understandable |
| semantic html | +0.68 | +0.01 | No | Robust |
| switch access | +0.47 | +0.03 | No | Operable |
| plain language | +0.70 | +0.07 | No | Understandable |
| alert dialog | +0.47 | +0.08 | No | Robust |
| screen reader | +0.50 | +0.10 | No | Perceivable |
| reduced motion | +0.69 | +0.10 | No | Operable |
| haptic feedback | +0.18 | +0.11 | No | Operable |
| tree grid | +0.76 | +0.16 | No | Robust |
| motion sensitivity | +0.57 | +0.18 | No | Operable |
| input purpose | +0.29 | +0.21 | No | Perceivable |
| aria attribute | −0.06 | +0.26 | No | Robust |
| focus management | +0.68 | +0.26 | No | Operable |
| color contrast | +0.25 | +0.28 | No | Perceivable |
| heading structure | +0.54 | +0.38 | No | Perceivable |
| cognitive load | +0.56 | +0.38 | No | Understandable |
| focus trap | +0.38 | +0.41 | No | Operable |
| alt text | +0.64 | +0.43 | No | Perceivable |
| audio description | +0.36 | +0.43 | No | Perceivable |
| screen magnifier | +0.50 | +0.47 | No | Perceivable |
| text resize | +0.78 | +0.51 | No | Perceivable |
| landmark region | +0.57 | +0.56 | No | Perceivable |
| **Strict decouple (majority vote):** 14/41; mean across 5 seeds = 42% | | | | |

Table 45: CRFM 117M C4-B 41-term breakdown (5-seed aggregate). 14 terms show majority-vote decoupling (mean per-seed fraction = 42%); high seed variance (seed 4: 73%, seed 3: 22%) reflects initialization sensitivity. Deepest aggregate decoupling for time limits ($\rho_{\text{late}} = -0.37$).

### A.1j.6: SmolLM3-3B C4-B (41 Terms)

| Term | $\rho_{\text{early}}$ | $\rho_{\text{late}}$ | Decoupled | Category |
|---|---|---|---|---|
| audio description | +0.20 | −1.00 | **Yes** | Perceivable |
| haptic feedback | +0.77 | −0.89 | **Yes** | Operable |
| eye tracking | +0.26 | −0.77 | **Yes** | Operable |
| landmark region | +0.26 | −0.77 | **Yes** | Perceivable |
| plain language | +0.26 | −0.77 | **Yes** | Understandable |
| focus management | +0.74 | −0.63 | **Yes** | Operable |
| heading structure | +0.32 | −0.63 | **Yes** | Perceivable |
| voice control | +0.60 | −0.60 | **Yes** | Operable |
| braille display | +0.95 | −0.40 | **Yes** | Perceivable |
| input purpose | +1.00 | −0.40 | **Yes** | Perceivable |
| keyboard shortcut | +0.11 | −0.40 | **Yes** | Operable |
| reflow content | +0.80 | −0.32 | **Yes** | Perceivable |
| tree grid | +0.32 | −0.32 | **Yes** | Robust |
| aria attribute | +0.40 | −0.26 | **Yes** | Robust |
| motion sensitivity | +0.26 | −0.26 | **Yes** | Operable |
| non-text content | +0.20 | −0.26 | **Yes** | Perceivable |
| orientation support | +0.77 | −0.26 | **Yes** | Perceivable |
| semantic html | +0.26 | −0.26 | **Yes** | Robust |
| color contrast | +0.60 | −0.20 | **Yes** | Perceivable |
| cognitive load | +0.95 | +0.00 | **Yes** | Understandable |
| screen reader | +0.32 | +0.00 | **Yes** | Perceivable |
| skip link | +0.80 | +0.00 | **Yes** | Operable |
| switch access | −0.80 | −1.00 | No | Operable |
| captions closed | −1.00 | −0.95 | No | Perceivable |
| live region | +0.00 | −0.80 | No | Robust |
| touch target size | −0.77 | −0.77 | No | Operable |
| keyboard navigation | −0.40 | −0.40 | No | Operable |
| text spacing | −0.21 | −0.40 | No | Perceivable |
| error identification | −0.20 | −0.32 | No | Understandable |
| sign language | −0.77 | −0.26 | No | Perceivable |
| alert dialog | −0.40 | −0.20 | No | Robust |
| skip navigation | −0.40 | −0.20 | No | Operable |
| text resize | −0.60 | −0.20 | No | Perceivable |
| alt text | +0.80 | +0.11 | No | Perceivable |
| focus indicator | −1.00 | +0.21 | No | Operable |
| high contrast | +0.32 | +0.40 | No | Perceivable |
| reduced motion | +0.26 | +0.45 | No | Operable |
| contrast ratio | −0.77 | +0.77 | No | Perceivable |
| focus trap | −0.77 | +0.77 | No | Operable |
| screen magnifier | +0.32 | +0.95 | No | Perceivable |
| **Strict decouple:** 22/40 (55%)[‡]; 1 term excluded (insufficient variance) | | | | |

Table 46: SmolLM3-3B C4-B 41-term breakdown (3440k steps). 22 terms show strict decoupling; deepest mean $\rho_{\text{late}} = -0.281$ across all terms indicates distributed redundancy regime. [‡]Due to left-censoring (earliest available checkpoint at step 40K is already post-coupling), decoupling is assessed as $\rho_{\text{late}} < \rho_{\text{early}}$ rather than strict $\rho_{\text{early}} > 0$, since the early-window sign is uninformative for this model.

## B  Appendix Figures

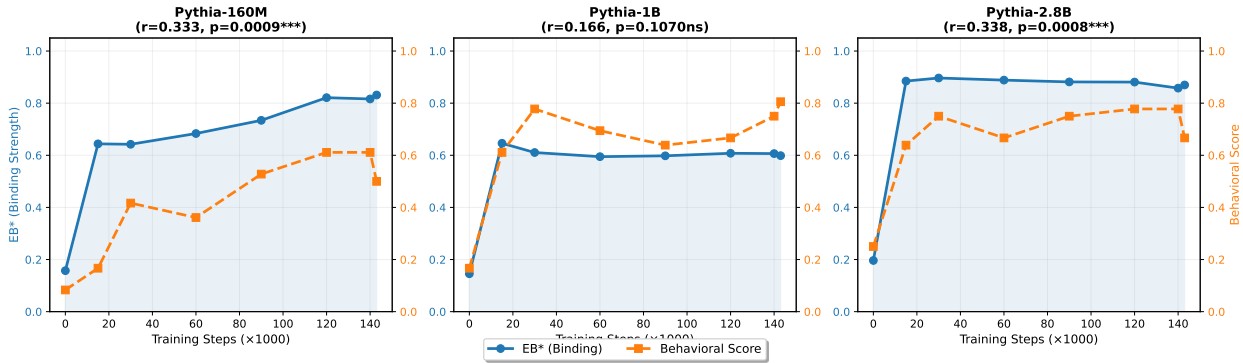

Figure 11: Emergence curves (behavior and EB*) across checkpoints for each model scale (3-term pilot data). 41-term canonical replication in Table 8 and Figure 6.

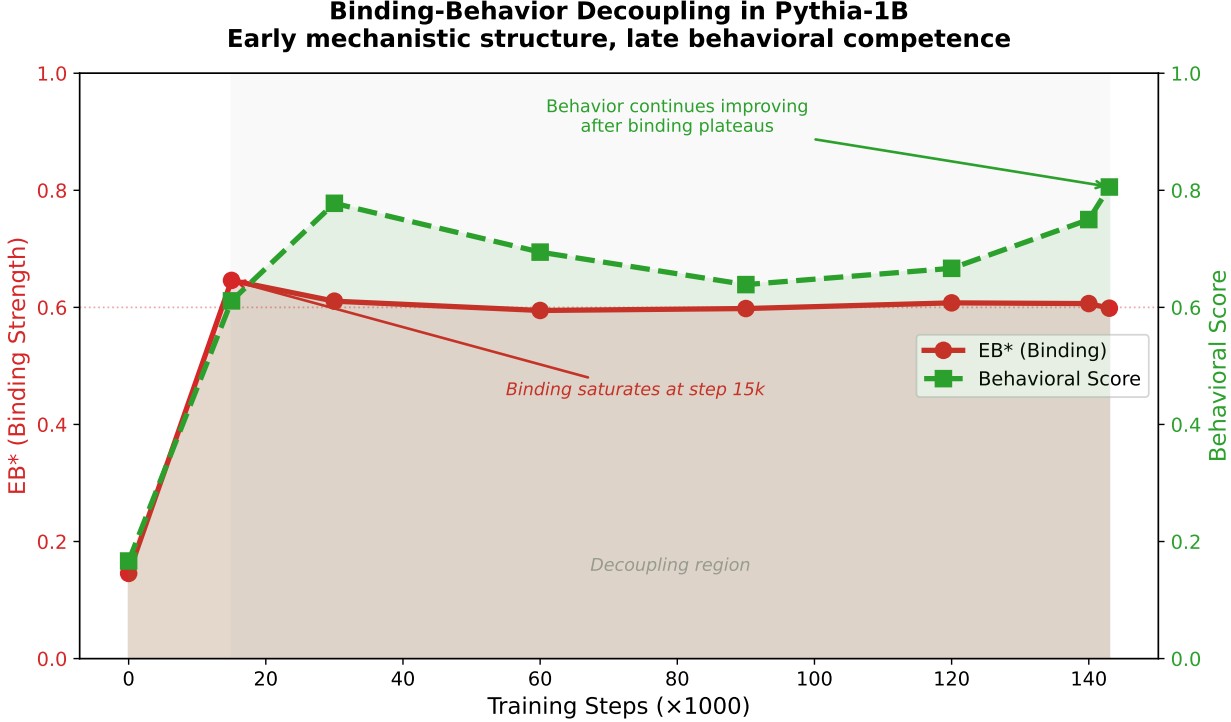

Figure 12: Decoupling at 1B scale (3-term pilot data): EB* saturates early while behavioral performance continues improving. 41-term canonical replication in Table 10.

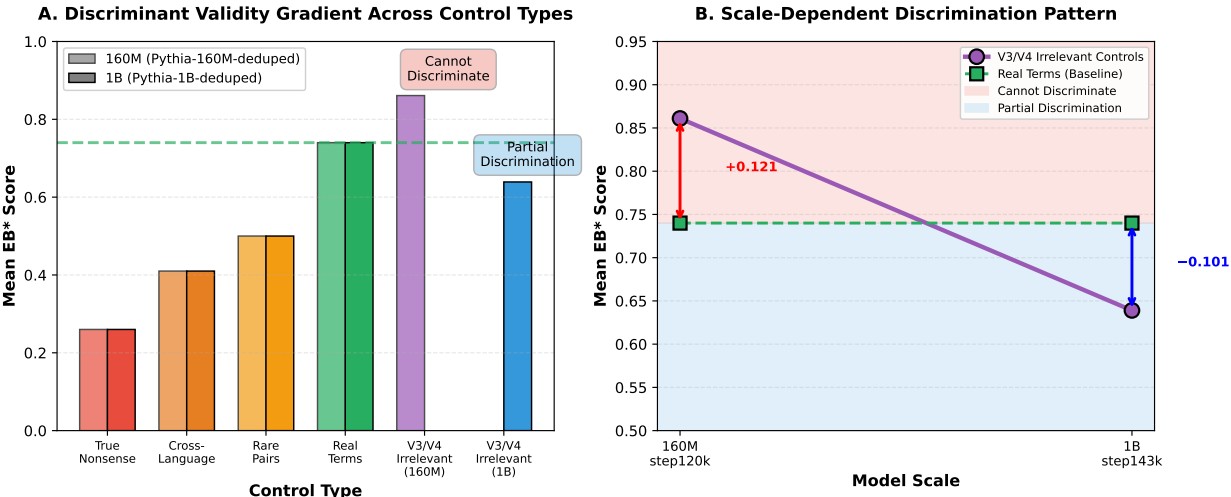

Figure 13: Discriminant validity gradient (full). Panel A: mean $EB^*$ across V2 control types (true nonsense, cross-language, rare pairs), real terms, and V3/V4 irrelevant controls at 160M and 1B. Panel B: scale-dependent discrimination trajectory: V3/V4 irrelevant terms exceed real term baseline by +0.121 at 160M (cannot discriminate) but fall below by −0.101 at 1B (partial discrimination).

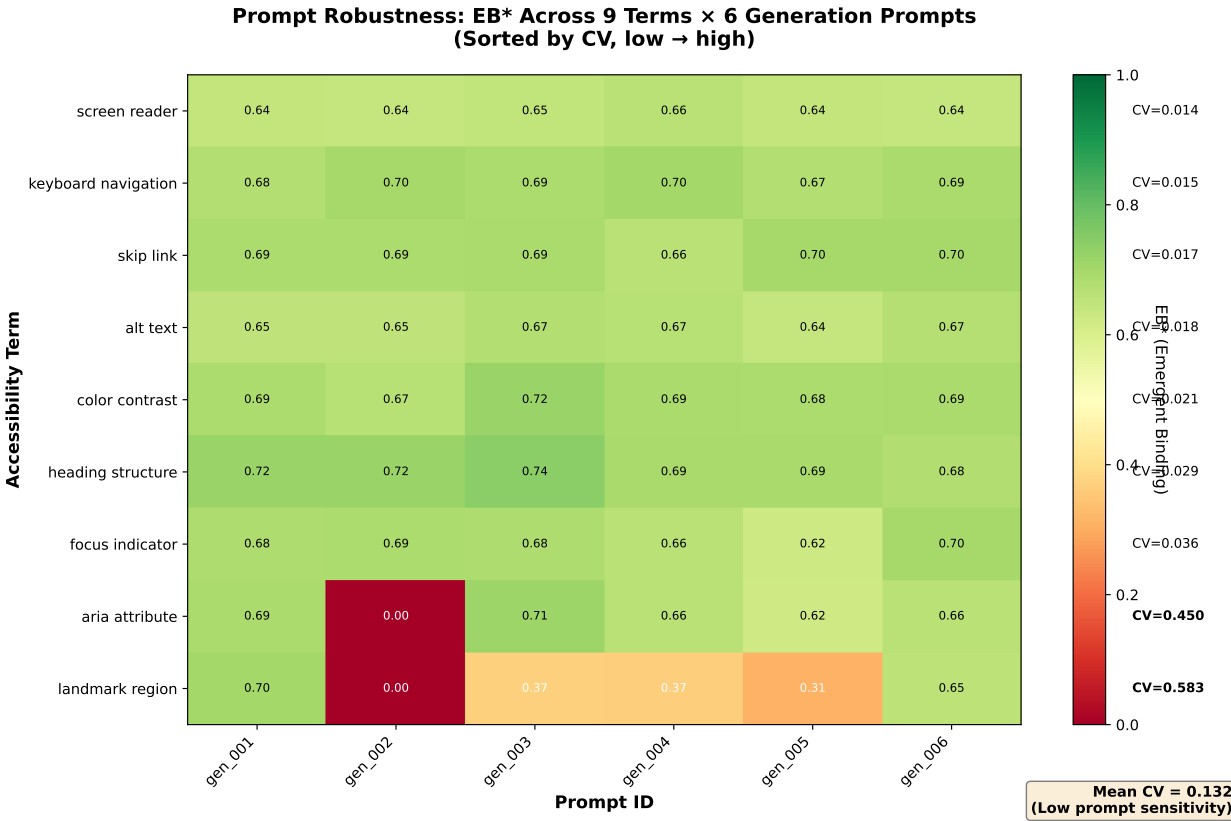

Figure 14: Prompt robustness heatmap. Mean EB* values across 9 terms × 6 generation prompt formats, sorted by coefficient of variation (CV). Color intensity indicates binding strength. 7/9 terms show CV < 0.05; "aria attribute" and "landmark region" are the high-variance outliers explained by the gen_002 prompt structure anomaly.

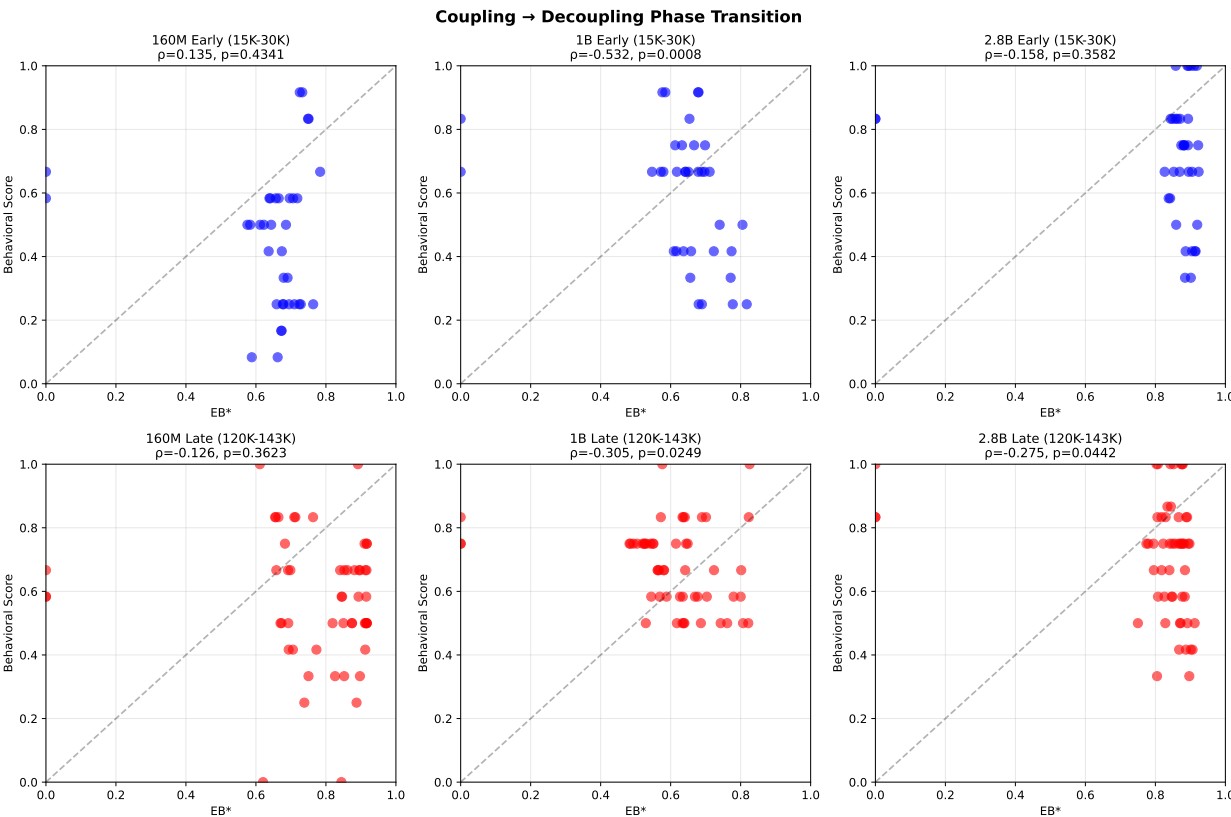

Figure 15: Phase transition scatter plots (appendix version). Six panels show EB* vs. behavioral score at early (blue, steps 15–30K) and late (red, steps 120–143K) checkpoints for all three model scales. Early checkpoints cluster above the diagonal; late checkpoints show decoupling at 1B and 2.8B.

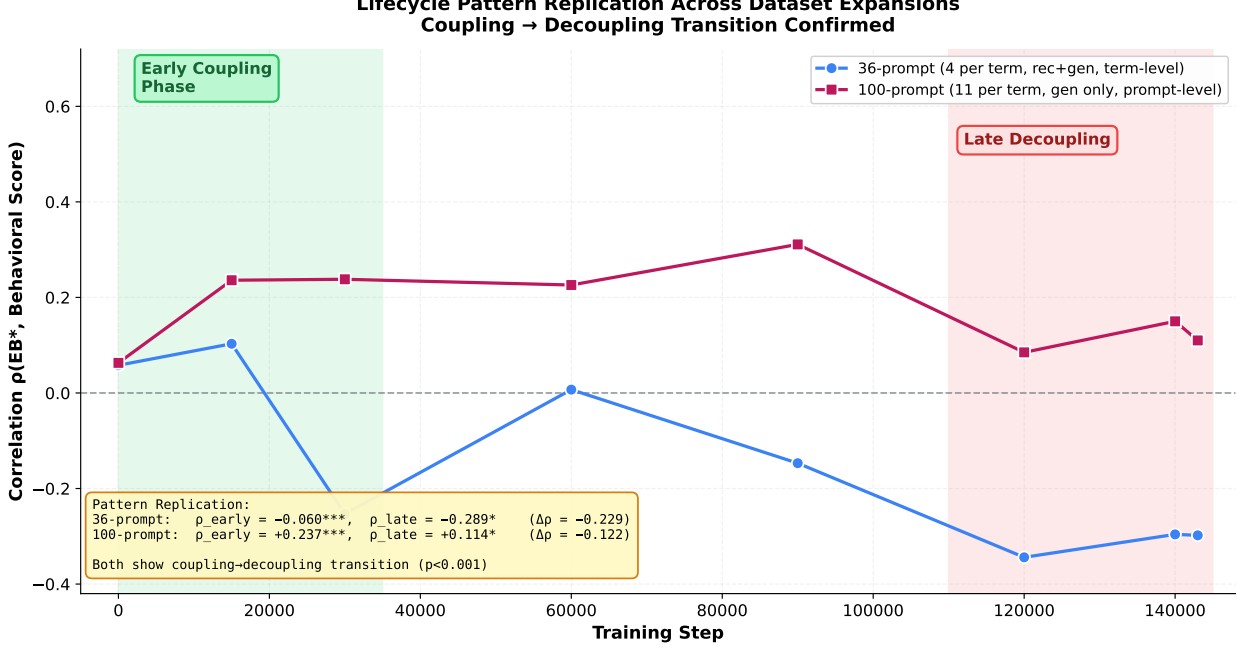

Figure 16: Lifecycle replication: 36-prompt (original) vs. 99-prompt (expanded) datasets. Both show the characteristic coupling→decoupling transition, validating that the pattern is robust to dataset composition and evaluation methodology.

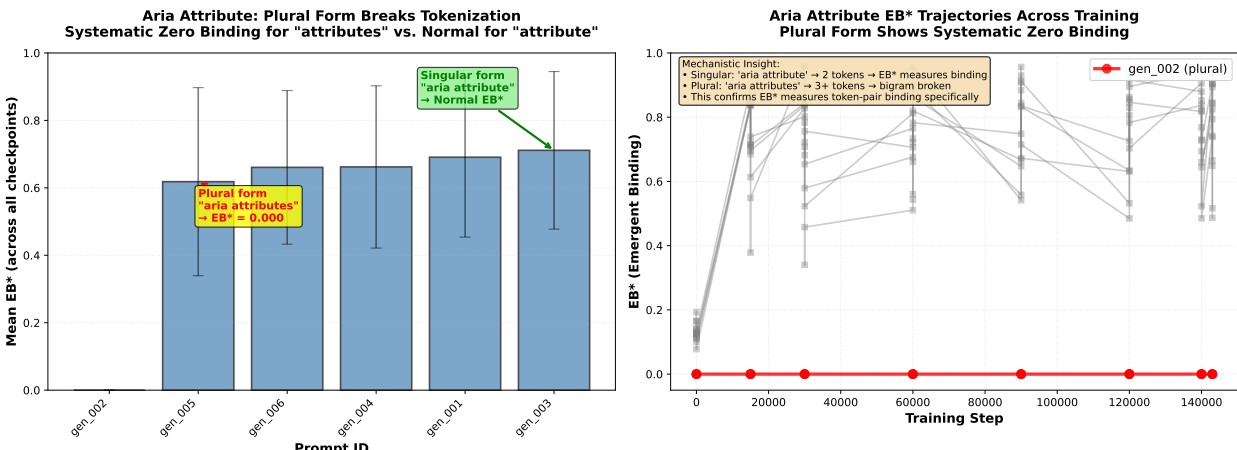

Figure 17: Aria attribute case study. Left: mean EB* by prompt ID, with gen_002 (sentence-final structure) producing EB* = 0.000 while other prompts (including other plural forms) maintain normal binding (0.62–0.71). Right: EB* trajectories across training checkpoints. This dissociation validates EB*'s construct validity as a token-pair-level metric.

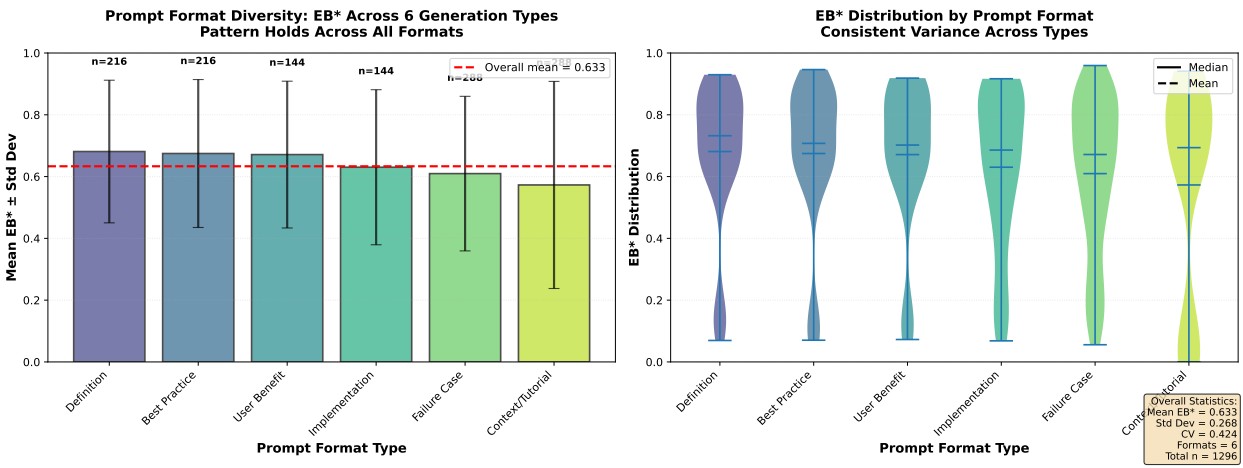

Figure 18: Format diversity analysis. Left: mean EB* by prompt format type (definition, user benefit, implementation, failure case, best practice, tutorial) with error bars. Right: EB* distributions via violin plots. All six generation formats produce comparable EB* distributions (0.57–0.68), confirming the lifecycle pattern is not format-dependent.

## C  Figure Index

Complete listing of all paper figures, their filenames, paper sections, and generation scripts:

| Fig | Filename | Section | Caption Summary |
|---|---|---|---|
| 1 | correlation_lifecycle.pdf,png | §4.3 | Mean Spearman $\rho$(EB*, Beh) at early/late checkpoints for Pythia scales (41-term) |
| 2 | phase_transition_scatter.pdf,png | §4.3 | Six-panel scatter (3 scales × 2 phases): EB* vs behavioral score |
| 3 | term_heterogeneity_2b8.pdf,png | §4.3 | Per-term EB* and behavioral trajectories at 2.8B scale |
| 4 | figure1_emergence_curves.pdf,png | §4.3 | Three-panel emergence curves showing EB* and behavioral score across training steps for 160M, 1B, and 2.8B models |
| 5 | figure4_1b_decoupling.pdf,png | §4.4 | Dual-axis: EB* saturates at step 15k while behavior rises through step 143k (1B) |
| 6 | prompt_robustness_heatmap.pdf,png | §4.2.2 | Mean EB* across 9 terms × 6 generation prompts, sorted by CV |
| 7 | lifecycle_comparison_36v100.pdf,png | §4.2.2 | 36-prompt vs 99-prompt datasets showing coupling→decoupling transition |
| 8 | aria_attribute_case_study.pdf,png | §4.2.2 | Aria attribute anomaly: gen_002 produces systematic zero binding |
| 9 | format_diversity_analysis.pdf,png | §4.2.2 | EB* by prompt format type (6 categories) with violin distributions |
| 10 | discriminant_validity_controls.pdf,png | §4.1 | Discriminant validity gradient: V2 controls → real terms → V3/V4 controls |
| 11 | c1b_forest_plot.pdf,png | §4.3.1 | C1-B forest plot: EB*-leads fraction per model (Wilson 95% CI) |
| 12 | c3_fewshot_unlockability.pdf,png | §4.5 | Panel A: Pythia 3×2 bars; Panel B: cross-model $\Delta$ for 11 model-checkpoint pairs |
| 13 | c5_crossarch_specificity.pdf,png | §4.6.1 | C5 cross-architecture causal specificity (7 models, CRFM error bars) |

Table 47: Complete figure index for the paper.

## D  Metric Definitions (Summary)

**Binding Strength Index (BSI).**  For a term $T$ with span positions $I_T = \{s_1, \ldots, s_n\}$, layer $\ell$, head $h$:

$$\text{BSI}(T, \ell, h) = \frac{1}{|\mathcal{P}|} \sum_{(i,j) \in \mathcal{P}} A_{\ell,h}[s_i, s_j], \qquad \mathcal{P} = \{(i,j) : s_i, s_j \in I_T, \ s_i > s_j\}.$$

**Effective Binding (EB).**

$$\text{EB}(T, \ell) = \max_h \text{BSI}(T, \ell, h) - \frac{1}{H} \sum_{h=1}^{H} \text{BSI}(T, \ell, h).$$

**Aggregate binding (EB*).**

$$\text{EB}^*(T) = \max_{\ell \in \mathcal{M}} \text{EB}(T, \ell).$$

**Repository pointer.**  Full code, prompts, and per-prompt outputs are publicly available at `https://github.com/RayoHQ/attention-binding-a11y`

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
