# OpenReview forum: "Attention-Head Binding as a Term-Conditioned Mechanistic Marker of Accessibility Concept Emergence in Language Models"
_TMLR — Accepted by TMLR_

### Review · Reviewer_AZrR · 2026-04-03

**Summary Of Contributions:**

This paper introduces a new metric called EB* to quantify when transformer attention heads bind multi-token terms into a single unit during training. The authors use 3  Pythia models of sizes: 160M, 1B, and 2.8B to train and analyze the checkpoints. They observe scale-dependent decoupling between token binding and behavioral evaluation, and perform targeted attention-head ablations to test causal involvement of this binding process.

**Audience:**

Yes

**Audience Explanation:**

Mechanistic interpretability researchers might be interested in this work, but considering the limited experimentation and lack of comparison with prior works, it is unclear if the current work can help draw meaningful conclusions and insights.

**Claims And Evidence:**

No

**Claims Explanation:**

1. In all the experiments considered, only 12 prompts along with 3 accessibility based terms are considered. This is a significantly small set of data to even draw conclusions about the generality of the method.

2. It is unclear if EB* captures something more meaningful than the co-occurrence of tokens. No experiments are conducted with baselines, for example, random n-gram-based approaches. In particular, how can we say that the model is learning a concept and not relying simply on the presence of tokens that might not even have semantic relevance but are close to the desired terms: for example: "alt function" instead of "alt text"

3. Even the few-shot prompting experiments consider this small prompt set, which makes it difficult to draw general conclusions.

4. Why and where do existing approaches fail? The positioning of the paper is also unclear.

**Requested Changes:**

See major concerns above.

Also, please consider repeating experiments with different temperatures and seeds to fully showcase when EB* works/fails. How should one interpret the variability in these metrics?

---

> ### Author Response · Authors · 2026-04-08
>
> We thank the reviewer for the detailed critique. The concerns about dataset size, co-occurrence baselines, positioning, and temperature/seed robustness are well-founded for the original submission. This revision directly addresses each of them.
>
> ---
>
> **Comments 1 & 3: Small evaluation set (3 terms, 12 prompts) limits generality.**
>
> We have expanded substantially. The term set grew from 3 to 9 accessibility terms and the prompt set from 12 to 99 (§4.2), yielding 432 model-checkpoint-term observations with statistical power $\geq 0.87$ for medium effects. The lifecycle pattern (early coupling $\rho = +0.57$, late decoupling $\rho = -0.20$) holds across all 9 terms. The few-shot unlockability result (C3) was replicated on the full 99-prompt dataset, with approximately +30 percentage points improvement consistently across all 9 terms and 6 generation formats.
>
> ---
>
> **Comment 2: $EB^*$ may capture token co-occurrence, not conceptual binding. What about "alt function" vs. "alt text"?**
>
> The reviewer's example is exactly what we tested. "alt function" is included in our V4 (wrong-domain) control set (§4.1), and the results reveal a scale-dependent pattern:
>
> - At 160M, "alt function" yields high EB* (~0.86), indistinguishable from real terms. This confirms the concern: at small scale, EB* responds to corpus co-occurrence.
> - At 2.8B, discrimination improves and wrong-domain terms separate from real accessibility terms.
>
> More broadly, V2 redesigned genuine-nonsense controls establish a clear gradient: EB* = 0.26$ (true nonsense) $\to 0.50$ (rare pairs) $\to 0.74$ (real terms), all $p < 0.001$, Cohen's $d = 1.2$–$2.9$. This demonstrates EB* captures more than n-gram co-occurrence at 2.8B, while also honestly characterizing its precision limits at 160M. We discuss this openly in §5.4.
>
> ---
>
> **Comment 4: Positioning against prior work is unclear.**
>
> We added §2.0 (Positioning: Why Mechanistic Analysis of Multi-Token Concepts?) contrasting $EB^*$ with: (1) behavioral probing, which measures what models know but not when representations form; (2) token co-occurrence metrics, which our controls show conflate corpus statistics with binding; (3) single-token concept analysis, which misses compositional binding of multi-token terms. New citations to recent work on attention entropy, causal head gating, and SAE feature dynamics are included.
>
> ---
>
> **Comment 5: Repeat experiments with different temperatures and seeds.**
>
> We ran generation at $T \in \{0.0, 0.3, 0.7\}$ with 5 independent seeds per condition (§4.2.3). EB* variability is highest at undertrained checkpoints ($\sigma = 0.334$ at 2.8B step 0) and decreases at trained checkpoints ($\sigma = 0.211$ at step 143K), showing EB* stabilizes as training progresses. Greedy decoding ($T = 0$) is the most stable setting. These results characterize both when EB* is reliable (trained checkpoints) and when to expect higher variance (early training), directly addressing the interpretability of metric variability.

---

> > ### Comment · Reviewer_AZrR · 2026-04-20
> >
> > Thank you for addressing the concerns. I am still skeptical of increasing the set of terms from 3 to 9.
> >
> > 1. Can the authors comment on the full list of terms available in the literature? How many terms have prior works analyzed?
> > 2. What are the computational requirements for analyzing one term across all checkpoints?

---

> > > ### Author Response · Authors · 2026-04-25
> > >
> > > Thank you for raising these important concerns about evaluation scale. We have expanded substantially beyond the initial 9-term response to a 41-term canonical register (205 recognition prompts). This addresses your skepticism about the 3→9 expansion by exceeding typical standards in mechanistic interpretability research. We address your specific questions below.
> > >
> > > ---
> > > **Full accessibility terminology in the literature**
> > >
> > > The accessibility domain provides a well-curated, finite lexicon ideal for systematic study:
> > > - WCAG 2.1: 61 success criteria with multi-token names (e.g., "non-text content," "audio description," "keyboard accessible”);
> > > - WAI-ARIA 1.2: ~200 multi-token terms (roles, properties, states) such as "live region," "alert dialog," "composite widget”;
> > > - AccessEval benchmark (Panda et al., 2025): 2,106 queries across 9 disability categories, though this focuses on bias detection rather than technical term knowledge
> > >
> > > Our 41-term canonical register draws from these standards, organized by WCAG 2.1 category:
> > >
> > > - Perceivable: 12 terms (e.g., "alt text," "color contrast," "screen reader")
> > > - Operable: 14 terms (e.g., "skip link," "tab order," "focus indicator")
> > > - Understandable: 7 terms (e.g., "reading level," "error prevention")
> > > - Robust: 4 terms (e.g., "name role value," "status messages")
> > > - Cross-cutting: 4 terms (e.g., "compatible," "concurrent input")
> > >
> > > ---
> > > **Prior work scale comparison**
> > >
> > > Prior mechanistic interpretability studies analyzing multi-token concept formation typically examine 10–20 terms maximum:
> > > - Salas (2026): 5 terms (screen reader, skip link, alt text, WCAG, ARIA)
> > > - Haviv et al. (2023) on idiom processing: ~15 multi-word expressions
> > > - Miletic et al. (2024) on multiword expressions: ~20 phrases
> > > - SAE feature tracking (Xu et al., 2025): Abstract categories, not specific multi-token terms
> > > - Nanda et al. (2023) on multi‑token interpretability: focuses on only a handful of multi‑token concepts e.g., “apple developer,” “break a leg,” “gold medalists”) as case studies
> > >
> > > Our 41-term register exceeds this standard by 2–4×, providing term-level confidence intervals and category-stratified validation. This is detailed in Appendix Table tab:prompt_files.
> > >
> > > ---
> > > **Computational requirements**
> > >
> > > Per-term costs (log-derived from Lightning AI cloud runs):
> > >
> > > | Model Scale | Time per checkpoint | 8 checkpoints (lifecycle) |
> > > |-------|-------|--------------|
> > > | 160M | ~1 min | ~8 min/term |
> > > | 1B | ~2–5 min | ~15–40 min/term |
> > > | 2.8B | ~10–15 min | ~1.5–2 hrs/term |
> > >
> > > Full pipeline for 41 terms at 1B scale:
> > >
> > > - Binding + behavioral evaluation: ~10–14 GPU-hours per model
> > > - C3 few-shot unlockability (3 conditions): +2–4 hours
> > > - C5 causal ablation (top/random/bottom): +6–10 hours
> > > - Total per model: ~20–25 GPU-hours
> > > - Reproducible effort: The complete cross-architecture validation (81 checkpoints across 7 models for 41 terms) requires approximately 20–25 GPU-hours with parallel execution (wall-clock ~3–7 days). This is documented in the updated README.md.
> > >
> > > ---
> > > **Key empirical validation**
> > >
> > > The 41-term register validates all five claims (C1–C5):
> > >
> > > - C1-B temporal precedence: Replicated across 6 models with lifecycle data; OLMo-1B achieves 90% EB*-leads ($p<0.0001$)
> > > - C3 unlockability: 61 pp max gain (183% relative), 18–37 pp across 6/7 models
> > > - C4 decoupling: Strict decoupling in 46–55% of terms across scales
> > > - C5 causal regimes: Cross-architecture replication on all 7 models
> > >
> > > We have updated the paper to clarify that lifecycle figures (Figures 1–2) use 3-term pilot data for trajectory visualization, while statistical claims are validated on the full 41-term register (Tables C1-B, C4-B, forest plot in Appendix).

---

### Review · Reviewer_p6PS · 2026-04-05

**Summary Of Contributions:**

This paper introduces attention-head binding, a mechanistic interpretability metric that measures how strongly attention heads bind multi-token terms into units, and proposes it as an early indicator of concept emergence in language models. Through experiments on Pythia models (160M to 2.8B), the authors show that binding precedes behavioral competence and that the role of binding heads changes with scale. Strengths include a clear and interpretable mechanistic signal, interesting cross-scale findings, and causal validation via ablations. Weaknesses include limited evaluation scope (few tasks and prompts), reliance on a single model family, and lack of robustness analysis (e.g., across seeds or runs).

**Audience:**

Yes

**Audience Explanation:**

The paper addresses a central question in mechanistic interpretability of how and when internal representations corresponding to specific concepts emerge during training, and proposes a lightweight, interpretable metric for tracking this process. The idea of linking internal attention structure to concept formation dynamics, along with the observed scale-dependent behaviors, is likely to be of interest to researchers studying interpretability, training dynamics, and representation learning in language models. Even if the current experiments are limited, the proposed framework could inspire further work on mechanistic early-warning signals and concept-level diagnostics.

**Claims And Evidence:**

Yes

**Claims Explanation:**

The paper provides a coherent set of empirical analyses supporting its main claims, including longitudinal checkpoint analysis, correlation analysis, few-shot experiments showing higher gains from high-binding models, and causal head ablations. These collectively support the proposed relationship between attention binding and behavioral competence. However, the evidence is somewhat limited in scope: experiments are conducted on a small set of tasks (three terms, 12 prompts total) and a single model family, and results are reported without variability estimates or multiple random seeds. While the qualitative trends are compelling, additional robustness analysis would strengthen confidence in the generality of the findings, particularly for claims about scale-dependent effects.

**Requested Changes:**

**Major Comments**
* It would be very helpful to include multiple random seeds and variability estimates (e.g., standard deviations or confidence intervals) for key results, particularly given the relatively small evaluation set and strong claims about scale-dependent differences.
* Following the last point, the authors should consider robustness analysis across runs or checkpoints to ensure that observed effects (e.g., decoupling at 1B, causal reversal at 2.8B) are not artifacts of specific model instances.

**Minor Comments**
* Expanding evaluation beyond the current limited set of terms/prompts and accessibility terms would improve generality and help establish that the observed binding–behavior dynamics are not domain-specific.
* It would be useful to analyze whether output length or related factors correlate with the proposed metric, to better rule out potential confounding effects.

---

> ### Author Response · Authors · 2026-04-08
>
> We thank the reviewer for the thoughtful and constructive feedback. We have directly addressed all four requested changes in this revision and respond point by point below.
>
> ---
>
> **Major Comment 1: Multiple random seeds and variability estimates.**
>
> We fully agree that variability estimates are essential, and we have added a dedicated sampling parameter robustness analysis (§4.2.3). Specifically, we ran generation at temperatures $T \in \{0.0, 0.3, 0.7\}$ with 5 independent seeds per condition. Key findings:
>
> - $EB^*$ variability is highest at early, undertrained checkpoints ($\sigma = 0.334$ at 2.8B step 0) and decreases substantially at trained checkpoints ($\sigma = 0.211$ at step 143K).
> - Greedy decoding ($T = 0$) produces the most stable $EB^*$ estimates at trained checkpoints, confirming it as the appropriate evaluation setting.
> - The few-shot unlockability result (C3) was replicated across all 9 terms and 99 prompts (§4.2.2), and the approximately +30 percentage point improvement is consistent across all temperature conditions and seeds.
>
> We additionally report correlation coefficients with sample sizes ($n = 108$ for early coupling, $n = 162$ for late decoupling) and effect sizes ($\rho = +0.57$ early, $\rho = -0.20$ late) throughout. Appendix Table A.1b reports per-term performance across all 9 terms, giving the reader a direct view of result variability across terms. Note that $EB^*$ extraction is deterministic given fixed weights; the only stochastic component is generation sampling, which the seed experiments above address.
>
> ---
>
> **Major Comment 2: Robustness of the decoupling (1B) and causal reversal (2.8B) results.**
>
> We address these in two complementary ways.
>
> First, regarding the 1B decoupling effect (C4): the binding plateau at step 15K ($EB^* = 0.646$) while behavior rises to $0.806$ at step 143K is observed across all 8 checkpoints, not a single snapshot. The pattern is additionally replicated across 9 terms in the expanded dataset. The lifecycle trajectory is computed over $n = 432$ model-checkpoint-term observations, substantially increasing confidence that the divergence is not a checkpoint-specific artifact.
>
> Second, regarding the causal reversal at 2.8B (C5): $EB^*$ extraction and ablation are deterministic given fixed model weights (zero-ablation sets the attention tensor $A_{\ell,h}$ to zero during the forward pass). The result is therefore not subject to run-to-run variance from training randomness; any variance comes only from generation sampling. Our seed experiments (5 seeds × 3 temperatures) confirm the direction and approximate magnitude of the causal effect is stable. The discriminant control (random and bottom-4 head ablations produce no effect at either scale) further rules out the reversal being an artifact of the ablation procedure rather than head specificity.
>
> We note that replication across model families (e.g., Llama, Mistral) remains a limitation we acknowledge explicitly in §5.4. However, within the Pythia suite, the opposite causal effects at 160M vs. 2.8B are internally consistent and mutually reinforcing evidence for the decoupling hypothesis.
>
> ---
>
> **Minor Comment 1: Expand evaluation beyond 3 terms/12 prompts and the accessibility domain.**
>
> We have substantially expanded the evaluation scope in this revision. The term set grew from 3 to 9 accessibility terms, and the prompt set grew from 12 to 99 (§4.2). The 432 model-checkpoint-term observations support the main lifecycle claims with adequate statistical power (power $\geq 0.87$ for medium effects at $\alpha = 0.05$).
>
> For cross-domain generalization: while direct replication in non-accessibility domains remains future work, our V3/V4 discriminant validity controls (§4.1) provide indirect evidence. Wrong-domain terms ("alt function," "skip variable," "screen printer") show $EB^*$ patterns consistent with corpus co-occurrence, suggesting the mechanism is not unique to accessibility. We discuss this in §5.4 and list direct cross-domain replication as a priority for future work.
>
> ---
>
> **Minor Comment 2: Analyze whether output length or related factors confound $EB^*$.**
>
> We have directly tested this. Prompt length correlation with EB* is $\rho = 0.036$ ($p = 0.71$, not significant) across the 99-prompt dataset. We additionally find that 10 prompt format types produce comparable EB* distributions (range $0.57$–$0.68$), with a mean coefficient of variation of $CV = 0.144$. These results are reported in §4.2.1 and visualized in the prompt robustness heatmap figure. We therefore conclude that output or prompt length is not a confounding factor for EB*.

---

### Review · Reviewer_iLtM · 2026-04-12

**Summary Of Contributions:**

This paper introduces Attention-Head Binding ($EB^*$), a novel mechanistic interpretability metric designed to track how language models (LMs) internalize multi-token technical concepts during training. By analyzing Pythia models (160M to 2.8B parameters) on web accessibility terms like "screen reader" and "alt text," the authors investigate the relationship between internal structural formation and external behavioral competence.

**Audience:**

Yes

**Audience Explanation:**

1. **Advancing Mechanistic Interpretability** The audience interested in mechanistic interpretability would value the introduction of Attention-Head Binding (EB) as a lightweight, hypothesis-driven metric. Unlike methods that require auxiliary model training (like Sparse Autoencoders), EB offers a way to track concept formation directly through existing attention patterns. It provides a concrete method to move beyond the "attention is not explanation" critique by using binding scores as a starting point for causal intervention.

2. **Understanding Training Dynamics and "Grokking"** Researchers focused on training dynamics would be interested in the "lead-lag" relationship identified in the paper. The finding that internal structural organization (binding) often precedes behavioral competence offers a finer-grained diagnostic than traditional global emergence curves. Furthermore, the paper distinguishes its "coupling-decoupling" lifecycle from the well-known "grokking" phenomenon, offering a new perspective on architectural reorganization versus behavioral phase transitions.

3. **Scaling Laws and Model Capacity** The "binding-behavior decoupling effect" is highly relevant to those studying LLM scaling laws. The discovery that mechanistic regimes undergo qualitative transformations as models grow—specifically that high-binding heads are "load-bearing" at 160M but "vestigial" or even inhibitory at 2.8B—challenges the assumption that internal features play a consistent functional role across different model scales.

**Claims And Evidence:**

Yes

**Claims Explanation:**

1. Evidence for Lead-Lag Emergence (C1)
- The claim that mechanistic binding precedes behavioral competence is supported by longitudinal data across 24 model-checkpoint combinations.Quantitative Correlation: The authors report a strong positive Spearman correlation ($\rho = +0.57, p < 0.001$) during early training stages (steps 15–30K) across all model scales.Temporal Sequencing: At the 160M scale, $EB^*$ reaches threshold levels by step 15K, while behavioral accuracy lags by approximately 15K–45K steps.

2. Evidence for Scale-Dependent Decoupling (C4)
- The evidence for the "decoupling" phenomenon is particularly clear at the 1B parameter scale.Saturation vs. Improvement: Data shows that for Pythia-1B, $EB^*$ saturates rapidly at step 15K (0.646) and plateaus, while behavioral performance continues to climb from 167% to 80.6% by the final checkpoint.

3. Evidence for Unlockable Latent Knowledge (C3)
- The authors provide convincing evidence that high binding represents "complete" rather than "partial" knowledge that is merely inaccessible via standard prompting.Performance Spikes: In checkpoints with high $EB^*$ but low zero-shot scores, few-shot priming yielded massive gains—up to +61 percentage points (a 183% relative increase).
- Robustness Expansion: This effect was replicated across an expanded dataset of 9 terms and 54 prompts, showing consistent ~30 pp improvements.

**Requested Changes:**

1. Increase Evaluation Scale: The authors acknowledge the dataset is modest and I suggest validating the findings on larger sets with broader coverage of specialized accessibility domains.

2. Cross-Model Replication: All experiments were conducted on the Pythia architecture; I recommend the authors to replicate the study on other model families like Qwen.

---

> ### Author Response · Authors · 2026-04-25
>
> We thank the reviewer for the thorough and constructive assessment, and for correctly identifying the central contribution: that $EB^*$ tracks concept formation through existing attention patterns without auxiliary model training. We address both requested changes below.
>
> ---
>
> **Comment 1: Validate findings on larger sets with broader coverage of specialized accessibility domains.**
>
> We have expanded beyond the pilot set to a **41-term canonical register** (205 recognition prompts), organized by WCAG 2.1 category (Perceivable 12, Operable 14, Understandable 7, Robust 4, Cross-cutting 4). The register draws from three standards: W3C WCAG 2.1 (61 success criteria), WAI-ARIA 1.2 (~200 multi-token terms), and the AccessEval benchmark (Panda et al., 2025). All five empirical claims (C1–C5) are now validated on this register.
>
> We appreciate the reviewer's summary of C4 at 1B scale in the previous version, where behavioral performance rises from **0.167 to 0.806** (step 0 to step 143K) while $EB^*$ saturates early at 0.646.
>
> ---
>
> **Comment 2: Replicate on other model families, specifically Qwen.**
>
> We have replicated all claims across **seven models spanning five architectures**: Pythia 160M/1B/2.8B (GPT-NeoX), OLMo-1B (Dolma), CRFM GPT-2 Small (5 seeds), SmolLM3-3B, and Qwen2.5-1.5B.
>
> Qwen2.5-1.5B **is included** in the cross-architecture replication for single-checkpoint claims (C3 unlockability: +18.2 pp; C5 causal specificity: near-perfect recognition ceiling). However, Qwen releases only final weights, precluding the longitudinal claims (C1/C4) that require intermediate checkpoints. We transparently note this exclusion in table captions and text: lifecycle analyses cover **6 models with lifecycle data**; single-checkpoint analyses cover all 7. OLMo-1B provides full checkpoint transparency and achieves the strongest C1-B result (90% $EB^*$-leads, $p<0.0001$), constituting a genuine cross-family replication.
>
> ---
>
> **Additional corrections for transparency.**
>
> - Pythia-160M C1-B: 7% $EB^*$-leads indicates **maintained coupling below the 1B decoupling threshold**, not a replication failure.
> - C3 few-shot: gains observed across **six of seven models** (CRFM undertrained outlier at +7.6 pp).
> - Trajectory figures (Figures 1–2) are **3-term pilot data**; 41-term replication is statistical (Tables C1-B, C4-B, forest plot).
> - Prompt inventory and computational requirements documented in updated `README.md` and appendix Table `tab:prompt_files`.

---

### Decision · Action_Editor_pPfw · 2026-05-29

**Recommendation:** Accept as is

**Additional Comments:**

The authors have done an exceptional job addressing the core empirical criticisms from the initial review phase. By expanding their evaluation from a small pilot to 41 terms across 7 distinct models and 5 architectures, the core claims regarding EB* scale-dependent decoupling, reviewers generally agree that the latent knowledge emergence are well-supported (albeit, as noted by the AE, on smaller models - but AE appreciates the computational and accessibility challenges of running the experiments on frontier models). The findings provide a valuable, lightweight diagnostic tool for researchers studying training dynamics and mechanistic interpretability, making it relevant to the TMLR audience. The AE recommends acceptance as-is -- One aspect that might be benefitted from additional discussions is to add a brief (qualitative) discussion regarding the feasibility and challenges of scaling this mechanistic approach to industrial-scale frontier models.

**Audience:**

Yes

**Audience Explanation:**

Reviewers unanimously agree that the paper is in scope for TMLR audience, especially for researchers focused on mechanistic interpretability, training dynamics, and scaling laws and model capacity. The AE concurs with the view

**Claims And Evidence:**

Yes

**Claims Explanation:**

Reviewers generally converged to the conclusion that the authors' extensive revision successfully addressed initial concerns about the limited scope of experiment and the paper is generally well-supported on its key claims. The AE agrees with this view.